# Cryo-EM structure of the complete *E. coli* DNA gyrase nucleoprotein complex

Arnaud Vanden Broeck [1,2,3,4], Christophe Lotz [1,2,3,4], Julio Ortiz [1,2,3,4] & Valérie Lamour [1,2,3,4,5]*

DNA gyrase is an essential enzyme involved in the homeostatic control of DNA supercoiling and the target of successful antibacterial compounds. Despite extensive studies, a detailed architecture of the full-length DNA gyrase from the model organism *E. coli* is still missing. Herein, we report the complete structure of the *E. coli* DNA gyrase nucleoprotein complex trapped by the antibiotic gepotidacin, using phase-plate single-particle cryo-electron microscopy. Our data unveil the structural and spatial organization of the functional domains, their connections and the position of the conserved GyrA-box motif. The deconvolution of two states of the DNA-binding/cleavage domain provides a better understanding of the allosteric movements of the enzyme complex. The local atomic resolution in the DNA-bound area reaching up to 3.0 Å enables the identification of the antibiotic density. Altogether, this study paves the way for the cryo-EM determination of gyrase complexes with antibiotics and opens perspectives for targeting conformational intermediates.

[1] Department of Integrated Structural Biology, Institut de Génétique et de Biologie Moléculaire et Cellulaire (IGBMC), 1 Rue Laurent Fries, 67404 Illkirch Cedex, France. [2] Centre National de Recherche Scientifique (CNRS) UMR 7104, Illkirch, France. [3] Institut National de Santé et de Recherche Médicale (INSERM) U1258, Illkirch, France. [4] Université de Strasbourg, Illkirch, France. [5] Hôpitaux Universitaires de Strasbourg, 1 Place de l'Hôpital, 67091 Strasbourg Cedex, France. *email: lamourv@igbmc.fr

The type IIA DNA topoisomerases (Top2) are nano-machines that control DNA topology during multiple cellular processes such as replication, transcription and cell division[1–4]. These enzymes catalyze the transport of a DNA duplex through a double strand break to perform DNA relaxation, decatenation, and unknotting. DNA gyrase plays a vital role in the compaction of the bacterial genome and is the sole type II topoisomerase able to introduce negative supercoils into DNA, a reaction coupled to ATP hydrolysis[5].

DNA gyrase A and B subunits assemble into a $A_2B_2$ hetero-tetramer of approximatively 370 kDa forming three molecular interfaces called N-gate, DNA-gate and C-gate allowing DNA binding and strand passage[6,7]. The flexibility of this large enzyme constitutes a challenge for the structural study of the multiple conformations it adopts during the catalytic cycle. Until recently, only the structures of isolated domains with drugs were known, providing partial information on this modular enzyme and not accounting for the allosteric connections between the catalytic domains[8–11]. The architecture of a full-length DNA gyrase from *T. thermophilus* in complex with a 155 bp dsDNA and cipro-floxacin was first solved using cryo-electron microscopy (cryo-EM) at a resolution of 18 Å[12]. This first model provided a rationale for probing mechanistic questions based on biochemical, single molecule, and FRET techniques[13–16], but the low resolution model did not allow deconvolution of discreet conformations of the full-length enzyme, leaving many mechanistic questions unresolved. Furthermore, the atomic details of the structure and of the drug binding site in the context of the overall conformation of DNA gyrase were not available at this resolution. In addition, the information derived from the thermophilic homolog does not account for all the mechanistic and structural specificities of the *E. coli* DNA gyrase, the genetic model for which functional data have been accumulated over the past decades. In particular, the *E. coli* DNA gyrase GyrB subunit possesses a domain insertion of 170 amino acids, not found in the *T. thermophilus* enzyme, that has been shown to help coordinate communication between the different functional domains[17]. Deletion of this domain greatly reduces the ability of *E. coli* DNA gyrase to bind DNA and decreases its ATP hydrolysis and DNA negative supercoiling activity[17].

DNA gyrase is a prime target for catalytic inhibitors such as aminocoumarins, or 'poison' inhibitors of the cleavage complex,

such as quinolones[18–21]. More recently, novel bacterial topoisomerase inhibitors (NBTIs), have been developed that also target the DNA cleavage activity of gyrase, but with a mechanism and target site different from the quinolones[22]. These molecules and in particular gepotidacin, represent a promising alternative and are currently in clinical trial for the treatment of *bacterial* infections[23–25]. However, the use of any antibiotics may lead to the appearance of mutations and the development of resistant strains[26]. Our capacity to understand the molecular determinants of drug resistance depends in part on the availability of complete 3D architecture of DNA gyrase nucleoprotein complexes with drugs.

In this study, we reveal the complete molecular structure of the *E. coli* DNA gyrase bound to dsDNA and to the NBTI molecule gepotidacin using cryo-EM. The structure of the entire complex was solved at 6.6 Å resolution and two conformations of the DNA-binding/cleavage domain in closed and pre-opening states were solved at 4.0 and 4.6 Å resolution, respectively. These sub-nanometer resolution structures reveal in detail the structural and spatial organization of the different functional domains of *E. coli* gyrase and of their connections. In particular, they elucidate the position of the conserved GyrA-box motif responsible for DNA wrapping and allow the unambiguous identification of the gepotidacin molecule in the cryo-EM map. The deconvolution of distinct conformations of the DNA-binding/cleavage domain provides a better understanding of the allosteric movements that the enzyme conducts at the early steps of G-segment opening, stabilized by the gepotidacin molecule. Altogether this work paves the way for the structure determination of gyrase complexes with additional antibiotics using cryo-EM and the in-depth analysis of its allosteric regulation.

## Results

**Cryo-EM structure of the DNA gyrase nucleoprotein complex.** The *E. coli* GyrB and GyrA subunits were overexpressed and purified separately before assembly in stoichiometric amounts and incubation with a double-nicked 180 bp DNA (Supplementary Table 1). The holoenzyme was further purified by size-exclusion chromatography resulting in a single and homogenous complex (Fig. 1a and Supplementary Fig. 1a). DNA supercoiling and ATP hydrolysis assays showed that the $A_2B_2$ complex is fully

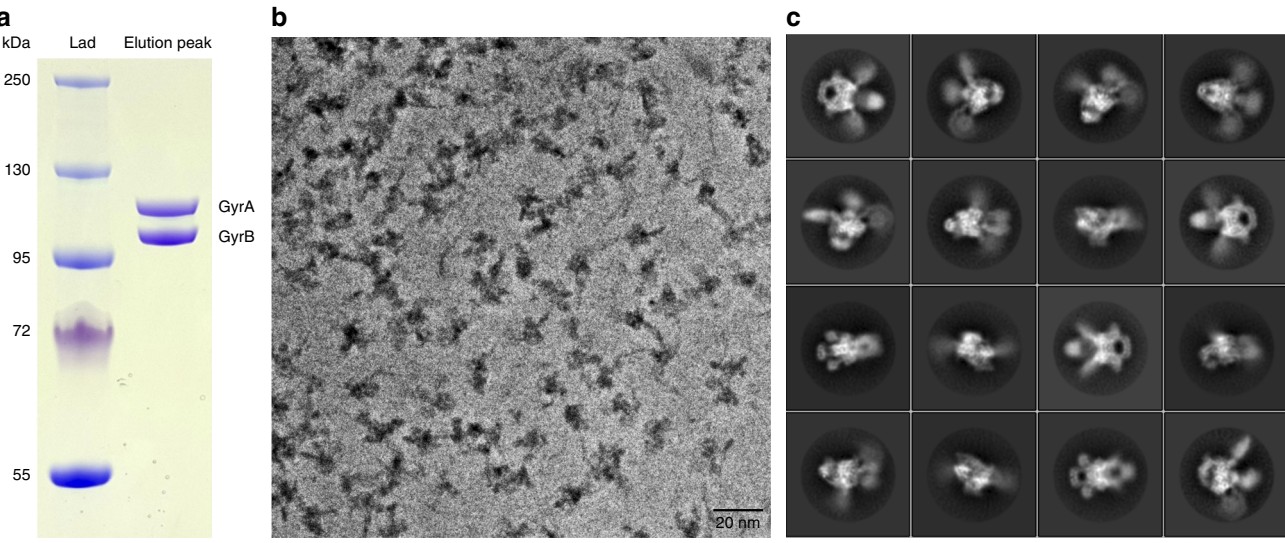

**Fig. 1** Sample preparation and image acquisition. **a** SDS-PAGE analysis of the reconstituted GyrA$_2$B$_2$ after size-exclusion chromatography. The source data are provided as a Source Data file. **b** A typical cryo-EM micrograph collected with a Gatan K2 Summit camera on a FEI Titan Krios microscope operated at 300 kV with Volta Phase Plate. The length of the scale bar is 20 nm. **c** Selection of 2D classes from reference-free 2D classification

active (Supplementary Figs. 1b and 9b). Finally, the gepotidacin molecule was added to the complex, which was further stabilized using ADPNP, a non-hydrolysable analog of ATP.

Images of the DNA gyrase complex were recorded using a Titan Krios with a Volta Phase Plate at a −500 nm defocus target, allowing a significant increase in contrast of the images and optimization of the image acquisition rate[27,28] (Fig. 1b). After gain correction, frame alignment and manual inspection, 8,701 micrographs were selected out of 11,833 initial movie frames coming from four datasets (Supplementary Table 2). Several rounds of 2D classification in RELION2[29,30] yielded class averages displaying high-resolution features such as the DNA wrapped around the β-pinwheels and defined alpha helices (Fig. 1c). An *ab-initio* 3D model was calculated with a final stack of 191,456 particles using cryoSPARC[31] (Supplementary Fig. 2). Multiple classification and refinement steps were then performed in RELION2 using global and local approaches to deconvolute different structures of the DNA-bound DNA gyrase (Supplementary Figs. 3–5). The overall DNA-bound DNA gyrase complex was solved at 6.6 Å using 94,633 particles. The estimation of the local resolution showed a broad spectrum from 5.0 Å on the DNA-binding/cleavage domain to 8.0 Å on the ATPase domains and β-pinwheels (Supplementary Fig. 4c).

Structures of the DNA-binding/cleavage domain were solved in two different states with different positioning of the GyrB TOPRIM insertion relative to the rest of the DNA-binding/cleavage domain. State 1 was solved at 4.0 Å using 60,548 particles and State 2 was solved at 4.6 Å using 53,655 particles (Supplementary Fig. 3). Local resolution estimation shows more stable regions reaching up to 3.0 and 4.0 Å for State 1 and State 2 structures, respectively, while the GyrB *E. coli* insertion seems more disordered (Supplementary Fig. 4c). To allow better positioning and refinement of the atomic models in the EM density of the overall structure, we also solved three additional structures with better defined densities for the ATPase domain and DNA-binding/cleavage domain (5.9 Å), the DNA-binding/cleavage domain without the TOPRIM insertion (4.0 Å) and the DNA-binding/cleavage domain with the C-terminal β-pinwheels wrapped by DNA (6.3 Å) (Supplementary Figs. 3–5 and Supplementary Table 2). The cryo-EM maps of individual regions at different resolutions were used to generate a composite EM map that reflects the flexibility of the complex, in particular in the ATPase domain and DNA-bound β-pinwheels (Fig. 2b, Supplementary Fig. 5 and Supplementary Movie 1).

**Analysis of the complete *E. coli* DNA gyrase architecture.** The atomic models of *E. coli* GyrA and GyrB domains, solved by X-ray crystallography[10,17,32], were combined to build and refine the complete atomic structure of the full-length *E. coli* DNA gyrase holoenzyme in complex with the DNA duplex in the cryo-EM maps (Fig. 2c). The continuous density of the ATPase and DNA-binding/cleavage domains EM map at 5.9 Å allowed us to build unambiguously the protein linkers between the C-terminal end of the transducer helices of the ATPase domain and the N-terminal end of the TOPRIM domain (Fig. 2d and Supplementary Fig. 5). The resulting model now shows clearly the details of the intertwined structure of the DNA gyrase hetero-tetramer (Fig. 2b, c). The dimeric ATPase domain is sitting in an orthogonal orientation above the DNA-binding/cleavage domain (~95°) as previously observed with the *T. thermophilus* DNA gyrase[12] (Supplementary Fig. 6a). The density of the flexible loops connecting the ATPase domain to the TOPRIM domain are well defined allowing to position these elements crossing on top of the G-segment in the DNA-bound gyrase conformation (Fig. 2d). In the DNA-free open structure of the *M. tuberculosis* DNA gyrase,

these linkers are laying apart on the surface of the TOPRIM domain[33].

We could also build the linker between the C-terminal end of the GyrA coiled-coil domains and the β-pinwheels on each side (Fig. 2e and Supplementary Fig. 5). The quality of the density allows to distinguish the upper from the lower face of the pinwheel disk where the N- and C-terminal ends are located. The tracing of the linker connecting to the N-terminal end of the pinwheel on blade 1 enabled the precise orientation of the crystal structure of the *E. coli* β-pinwheel[10] in the EM map (Figs. 2e and 3a). Consequently, the first contact between the G-segment and the β-pinwheel occurs at blade 3, then wraps around the β-pinwheel by contacting blade 4, 5, 6 and exits the β-pinwheel through contact with blade 1 containing the GyrA-box motif (Fig. 3a).

Other structural elements missing in crystal structures such as surface loops, α-helices of the DNA binding/cleavage and C-gate domains could be completed or corrected in the GyrB and GyrA subunits representing more than 5% of the total sequence (Supplementary Fig. 7 and Supplementary Table 3).

The overall structure is asymmetric, with the ATPase domain slightly bent (~10°) towards one of the β-pinwheels bringing it closer to a distance of 27 Å. The ATPase domain is sitting 11 Å away from the DNA-binding/cleavage domain (Supplementary Fig. 6). Each β-pinwheel resides on the same side as its respective GyrB subunit, with an approximate ~45° angle from the center of the DNA-binding/cleavage domain. This spatial arrangement of the β-pinwheel induces an overall ~150° bending of the DNA, as seen in previously reported crystal structures[22,34,35]. 130 bp of the double-stranded DNA out of 180 bp could be fitted and the corresponding base pairs were submitted in refinement in the cryo-EM map. The DNA duplex is chirally wrapped around the β-pinwheels orienting the T-segment towards the DNA-gate second groove formed by the TOPRIM-WHD and tower domains with a 60° angle (Supplementary Fig. 6b).

**Conserved structural elements in the transducer domain.** The transducer helices in the GyrB subunit are thought to be a pivotal structure for the allosteric changes following ATP hydrolysis[32,36]. These helices are sticking out of the transducer domain that forms a central cavity in the ATPase domain. Based on the first crystal structure of the ATPase domain[8], it was initially suggested that residue R286 in *E. coli* (*Ec*) points towards the cavity of the transducer domain and contributes to DNA capture and therefore is a key element of the enzyme's allostery[37]. However, further analysis of this area from the crystal structure of Brino et al. shows that *Ec* R286 seems engaged in a salt bridge with E264, which is also involved in a hydrogen bond network with R316[32] (Fig. 4b). R286 is replaced by a lysine in the *Thermus thermophilus* (*Tth*) enzyme and most organisms possess either an arginine or a lysine at this position (Supplementary Fig. 8). *Tth* K284 at the same position points toward the N-gate central cavity and is not engaged in an hydrogen bond network since E264 is replaced by A262 and R316 by a L314[38] (Fig. 4a). This organism therefore presents an alternative sequence to *E. coli* in this region with a conserved folding and accomplishing the same set of catalytic activities.

To probe the role of this position in the allosteric regulation of the holoenzyme, we compared the catalytic activities of point mutations of *Ec* R286 and *Tth* K284 (Fig. 4c, d). *Ec* R286 was mutated into a lysine to mimic the *Thermus* position and a glutamine to remove the charge as performed in the previous study of Tingley et al.[37]. The residue *Ec* E264 was also mutated into alanine to mimic the *Thermus* position. Inversely, *Tth* K284 was mutated into arginine or glutamine. The mutant proteins were

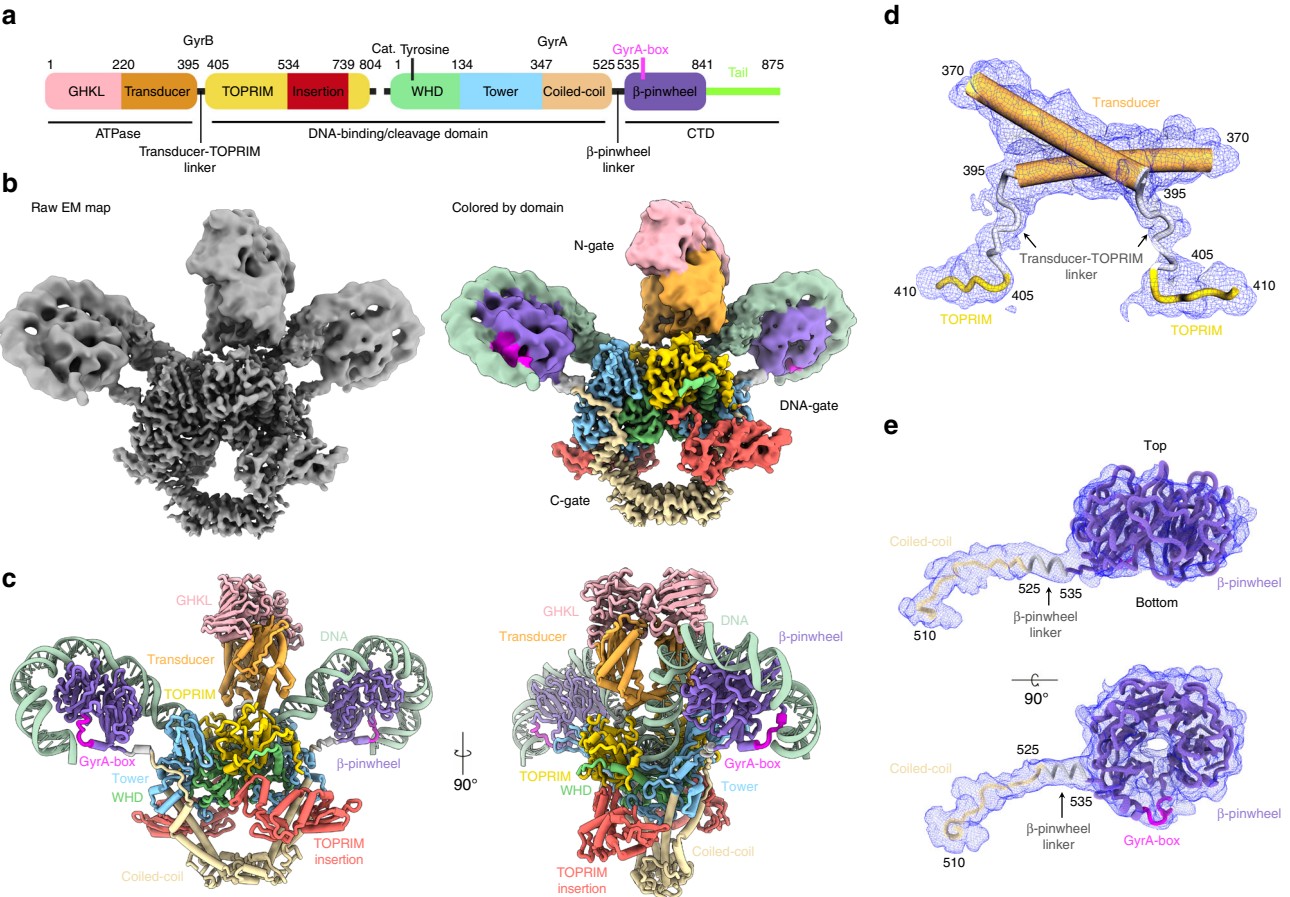

**Fig. 2** Cryo-EM 3D reconstruction and molecular model of the DNA gyrase complex. **a** Schematic representation of the GyrA and GyrB domains, the position of the conserved catalytic tyrosine in the DNA-binding/cleavage domain and of the GyrA-box are indicated. **b** Raw composite cryo-EM map of the full complex of DNA gyrase (left) and the same cryo-EM map colored by protein domains using the color code as in a (right). **c** Molecular structure of the hetero-tetrameric complex including the GyrA/GyrB subunits (same color code as in **a**) and 130 bp DNA (in pale green) that were built and refined in the cryo-EM maps. **d** Zoom on the transducer helices and the transducer-TOPRIM linkers built in the density (blue mesh) and connecting the N-gate to the DNA-gate. **e** Perpendicular views of the linker that connects the DNA-binding/cleavage domain to the bottom of the β-pinwheel. The identification of a clear density for the 10-AA linker ending with a short helix allowed to orient the β-pinwheel structure unambiguously in the map. This leads to the positioning of the GyrA-box (in magenta) at the exit of the DNA path around the β-pinwheel, in close contact with DNA

recombinantly produced with no detectable denaturation. Thermal denaturation experiments on the *E. coli* mutant proteins using differential scanning fluorimetry showed the same behavior as the WT protein except for the E264A that shows a decreased thermal stability (Supplementary Fig. 9a). The ATPase and DNA supercoiling activities of the *Tth* K284R and K284Q mutants remain in the same range as the WT at 37 °C (Fig. 4c and Supplemental Fig. 9b). The *Ec* R286K mutant exhibits slightly higher ATPase and DNA supercoiling activities compared to the WT suggesting either side chain is compatible with the catalytic activities in *E. coli* (Fig. 4d and Supplementary Fig. 9b). In contrast, the *Ec* R286Q and E264A mutants have a 6- and 8-fold reduced ATPase activity and present a 5- and 10-fold reduced negative supercoiling activity, respectively. The E264A mutation is particularly detrimental to the DNA gyrase activities. This could be explained by the fact that this residue forms hydrogen bonds with residues R286 and R316, which are both involved in the interaction network maintaining the structure of the transducer domain (Fig. 4b). The alanine mutation is likely to break these interactions and affect the rigidity of this structure element, as shown by the thermal denaturation assays (Supplementary Fig. 9a).

All together this suggests that the presence of the transducer interaction network in *E. coli* is important for ensuring allosteric coupling between ATP hydrolysis and DNA supercoiling activities.

**Deconvolution of the DNA-gate conformations**. The deconvolution of different particles population in the dataset has led to distinct 3D reconstructions of the DNA-binding/cleavage domain. The 4.0 Å (State 1) and 4.6 Å (State 2) structures of the DNA-binding/cleavage domain bound to the double-nicked G-segment reside in a configuration similar, but not identical, to the cleavage complex in presence of DNA and ciprofloxacin with a RMSD of 1.78 and 2.01 Å, respectively[12]. The overall RMSD between State 1 and State 2 conformations is of 2.2 Å. In state 2, very subtle changes occur in structural elements lining the G-segment groove compared to State 1. The 2 conformations mostly differ in the GyrB subunit, more particularly the TOPRIM insertion that rotates around the GyrA N-terminal arm with a 5.0 Å upward displacement (Fig. 5). This motion increases the distance between GyrA Tower and GyrB TOPRIM from 8 to 11 Å, inducing a slight shortening of the distance between the catalytic tyrosines from 26.4 to 25.6 Å and inducing the stretching of the G-segment by 2.5 Å in both directions (Fig. 5b, d and Supplementary Fig. 10).

State 1 can be described as a cleavage complex in a closed conformation of the DNA-gate, hereafter designated as closed. State 2 corresponds to an intermediate opening of the DNA-gate therefore referred as a pre-opening conformation, when compared with the larger opening that can be observed in the

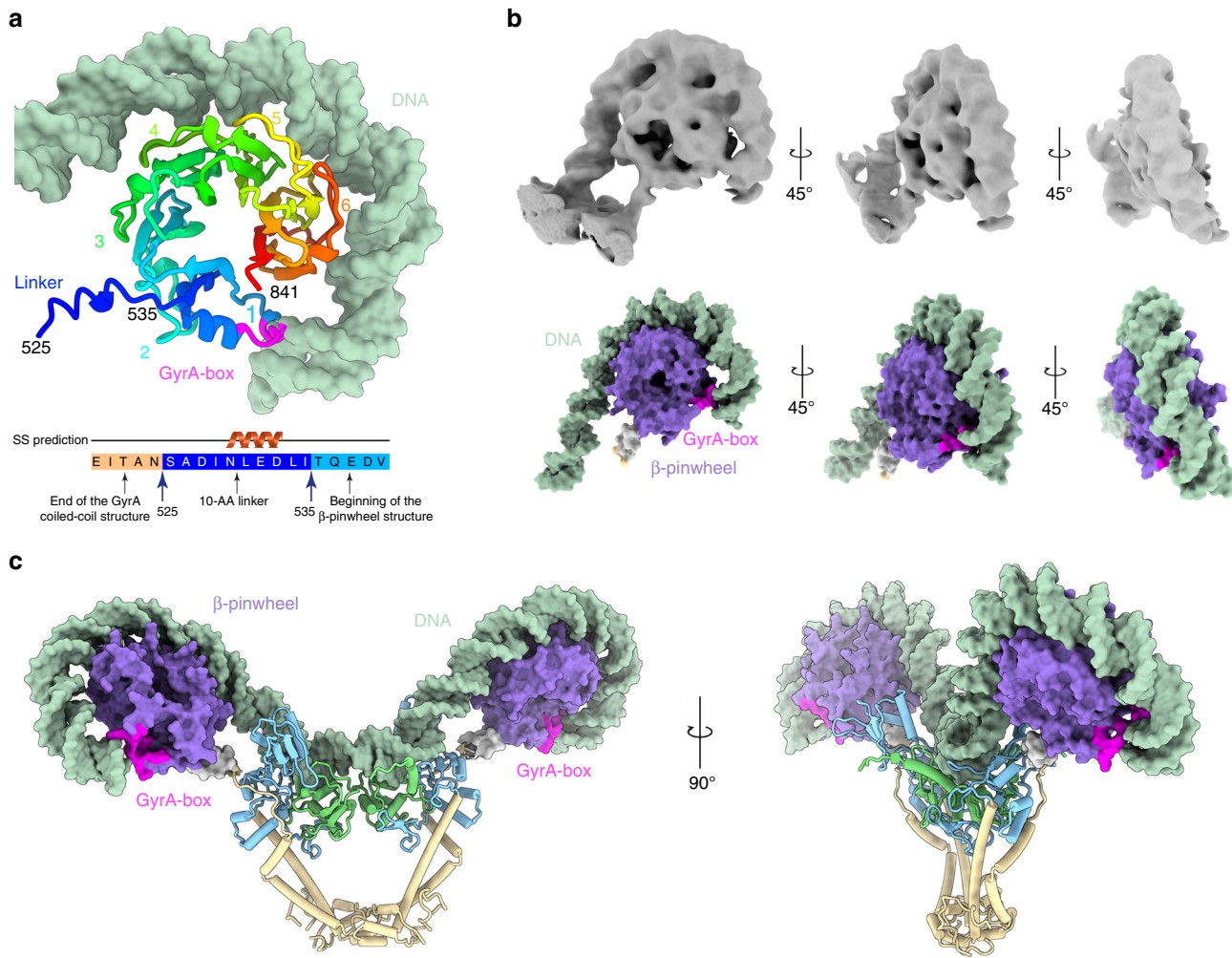

**Fig. 3** DNA Wrapping around the GyrA CTD β-pinwheel and GyrA-box structure. **a** Cartoon representation of the molecular structure of the GyrA β-pinwheel, rainbow colored from the N-terminal end in blue, to the C-terminal end in red. The GyrA-box (QRRGGKG) is colored in magenta and DNA in pale green. The β-pinwheel blades are numbered from 1 to 6. The first contact between DNA and the β-pinwheel occurs at blade 3, wraps around the disk by contacting blade 4, 5, 6 and exits the β-pinwheel through contact with blade 1. **b** Different views of the 6.3 Å cryo-EM map (in grey) zoomed on the β-pinwheel wrapped with DNA and the corresponding molecular models in surface representation (same color code as in Fig. 2). The slight superhelical structure of the pinwheel is clearly visible on the side view. **c** Overall view of the 130 bp DNA duplex wrapping around the two β-pinwheels of DNA gyrase. For clarity, only the GyrA subunits are displayed. The GyrA-box motifs of each β-pinwheel are located at the exit of the DNA path around the β-pinwheel and act as clamps stabilizing the DNA curvature

conformation adopted by the hsTop2β[39] (Supplementary Fig. 10d). State 1 and 2 correspond to a conformational oscillation before the opening of the DNA-gate (Supplementary Movie 2). This pre-opening transition is mainly undertaken by the insertion domain acting like an anvil which positions the TOPRIM domain in a configuration ready to perform the opening of the DNA-gate.

**Identification of the gepotidacin and ions in the EM map.** Despite the high flexibility of the complex, we were able to solve a structure of the gyrase DNA-binding/cleavage domain in the closed complex with DNA and gepotidacin at 4.0 Å resolution using a focused refinement strategy (Supplementary Fig. 3). Side chains of secondary structure elements and the DNA double helix are very well defined in this highly ordered region (Fig. 6a). The local resolution near the NBTI binding site was estimated at 3 Å, which allowed the identification of the gepotidacin density (Fig. 6a, b and Supplementary Fig. 4). The molecule intercalates in the DNA duplex between positions +2 and +3 forming hydrophobic interactions and pi-stacks with the protein and the

DNA, at the same location as previously observed for the NBTIs and crystallized with *S. aureus* DNA gyrase[22,25,40,41] (Supplementary Fig. 11a). The main difference in the binding pocket of *S. aureus* and *E. coli* gyrase lies in the subunit methionine residue (M75) that is replaced by an isoleucine residue in (I74) (Supplementary Fig. 11a). This position is part of a GyrA hydrophobic pocket that accommodates the pyranopyridine ring of the gepotidacin. Although the overall resolution of the pre-opening state is lower, a density for the gepotidacin molecule could be identified at a lower rmsd level, suggesting that the gepotidacin binding site is compatible with conformational fluctuations at the DNA-gate. Such flexibility of the DNA-binding/cleavage domain in presence of gepotidacin has also been observed in the *S. aureus* DNA crystal structures of DNA-binding/cleavage domains with an intact or doubly nicked DNA[25].

Further away along the DNA in the TOPRIM domain, a density could be observed in the 4 Å EM map at 6 Å rmsd at the cleavage reaction B-site that could correspond to the position of a magnesium ion (Supplementary Fig. 12). The density of the conserved aspartic and glutamic residues defining the B-site are

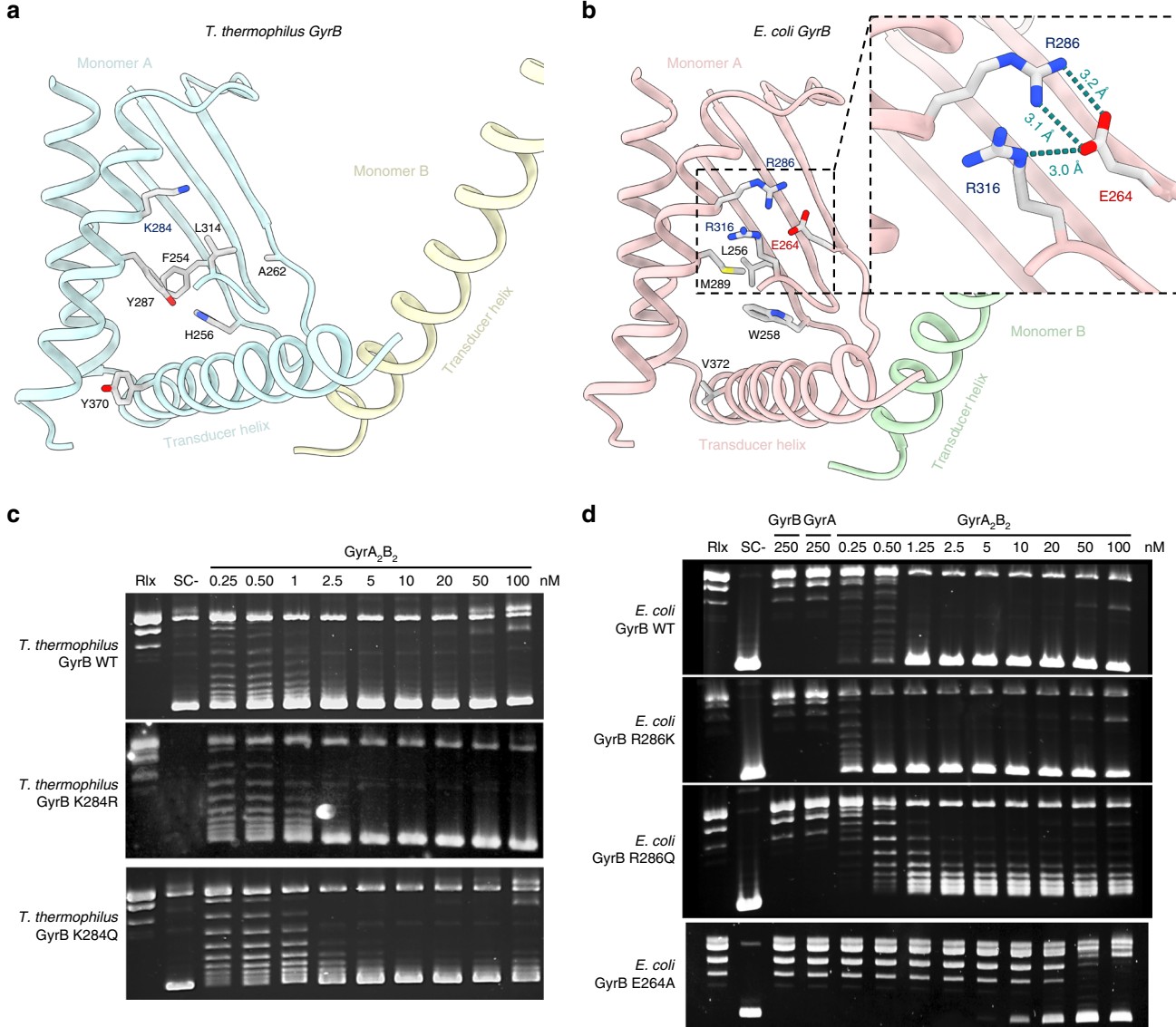

**Fig. 4** Transducer structural elements involved in allosteric regulation. **a** Cartoon representation of the *T. thermophilus* GyrB transducer domain. The K284 residue is not engaged in an interaction network. The transducer is mainly stabilized by a hydrophobic core (only side chains different in the two species are displayed) and a hydrogen bond involving H256 and Y287. **b** Cartoon representation of the *E. coli* GyrB transducer domain. The transducer domain is stabilized by a salt bridge involving R286 and E264 in interaction with R316, anchoring the beta-sheets to the alpha helix. **c** DNA negative supercoiling activities of wild-type, K284R and K284Q *T. thermophilus* DNA gyrase showing no effect of the mutations. Protein concentrations are indicated in nM holoenzyme. Negative and positive controls are shown as relaxed (Rlx) or negatively supercoiled DNA species (SC−), respectively. **d** DNA negative supercoiling activity of wild-type, R286K, R286Q, and E264A *E. coli* DNA gyrase showing no effect of R286K but an impaired activity of the R286Q and E264A mutants. The source data are provided as a Source Data file

however much weaker but it has now been observed in high-resolution cryo-EM maps that negatively charged groups yield weaker densities than positively charged groups[42]. Taken together, it is likely that the cryo-EM structure could contain the requisite metal ions needed by DNA gyrase to perform DNA cleavage.

## Discussion

Although the complex was formed with a 180 bp linear DNA template, only 130 bp could be built in the cryo-EM map, showing that the remaining 25 or 50 bp distributed on both sides of the G-segment are highly flexible. This was also observed in the *T. thermophilus* gyrase complex with DNA, where 130 bp out of 155 bp could be docked in the cryo-EM map albeit of a lower

resolution[12]. The spatial arrangement of the β-pinwheels induces an overall ~150° bending of the G-segment in the *E. coli* gyrase. The 130 bp DNA duplex is chirally wrapped around the β-pinwheel orienting the T-segment in the DNA-gate groove formed by the TOPRIM and tower domains to form a 60° angle positive crossover when T-segment path is extrapolated. The combination of the high bending angle of the G-segment (~150°) with the peculiar orientation of the β-pinwheels positions the T-segment to access the DNA-gate with an almost null or negative angle of ~−10° (Supplementary Fig. 6b). As a consequence, this configuration seems unfavorable to T-segment strand passage and would require a structural rearrangement of the β-pinwheel to proceed further. The *E. coli* structure is reminiscent of a state immediately following G-segment binding and wrapping, a conformational intermediate that could have been trapped by

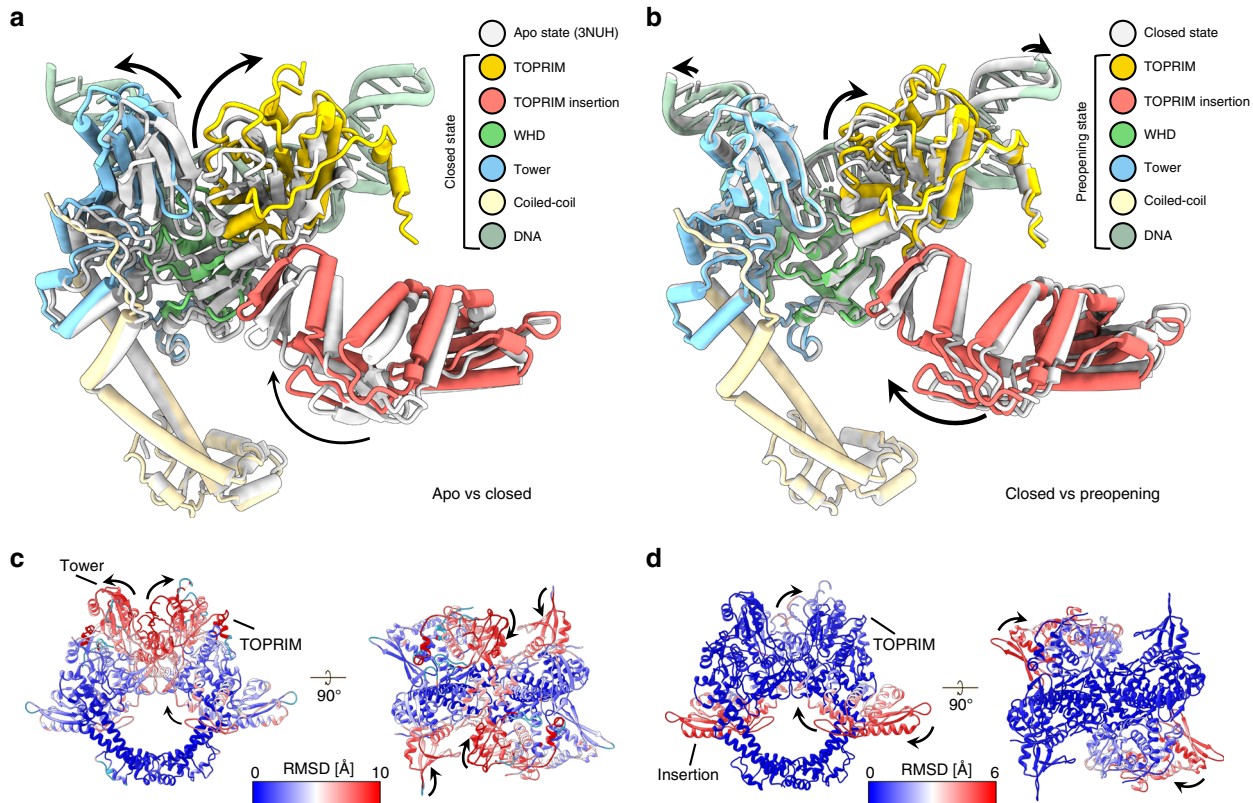

**Fig. 5** Conformational changes associated with G-segment binding, cleavage and opening. **a** Superimposition of the cleavage complex in a closed state (this study) to the DNA-free DNA-binding/cleavage domain (*apo* state, PDB ID 3NUH). Only one monomer of GyrB and GyrA is displayed for clarity. The superimposition shows that the majority of the movements are performed by the TOPRIM and Tower domains upon DNA binding and cleavage. **b** Superimposition of the cleavage complex in the pre-opening state to the closed state. The superimposition shows the upward movement of the TOPRIM domain while the tower domain remains fixed. The TOPRIM insertion domain pivots around the N-terminal GyrA arm preceding opening of the G-segment after cleavage. **c** RMSD analysis of the DNA-binding/cleavage domain in *apo* state and cleavage complex in closed conformation. **d** RMSD analysis of the cleavage complex in closed conformation to the pre-opening conformation

gepotidacin. Interestingly, a recent single-molecule study identified a Ω conformational state after G-segment binding, possibly similar to the *E. coli* complex and preceding the α state in the catalytic cycle, a conformation favorable to the T-segment strand passage[13]. This transition would require a different orientation of the pinwheels which motion might be controlled and regulated by the acidic CTD tail in contact with the DNA-gate as previously suggested[43].

During T-segment strand passage, the DNA-binding/cleavage domain needs to undergo several conformational changes. Crystal structures of DNA-binding/cleavage domains in different states have led to propose that the DNA-gate opening is achieved by a sliding and swiveling motion of the two halves against each other, breaking the G-segment axis[2,3,39,44]. A recent structure of the hsTop2β DNA-binding/cleavage domain was solved in an open conformation, revealing a funnel-shaped channel ready for the entry of T-segment[40]. Besides, MD simulation of the T-segment crossing through the open DNA-gate suggested the presence of a conformation where the central cavity is widened and flattened, a state previously observed in a crystal structure of the *B. subtilis* DNA gyrase[45]. Consequently, the high degree of structural conservation of the bacterial and the human Top2β suggests that this mechanism of sliding and swiveling can be extended to the bacterial Topo IIA, and thus to DNA gyrase. In this context, the pre-opening conformation of the cleavage complex represents one of the discreet steps that DNA gyrase undergoes to open the DNA-gate (Supplementary Movie 2). In *E. coli*, this step might be further controlled by the presence of the GyrB insertion domain.

Deletion of the insertion was shown to greatly reduce the DNA binding, supercoiling and DNA-stimulated ATPase activities of *E. coli* DNA gyrase[17]. The subtle swiveling movement of the GyrB insertion domain around the N-terminal arm of GyrA leads to a controlled stretching of the G-segment by 5 Å and spatial separation of the TOPRIM and Tower domains, a necessary step for the formation of the groove that will accommodate the T-segment. In the light of our structural study, the insertion in the TOPRIM domain may presumably act as a steric counterweight amplifying the movements of the DNA-gate.

The DNA-gate opening is coupled to ATP hydrolysis through the transducer region whose cavity is thought to accommodate a DNA T-segment, as evidenced by recent structural data on a bacterial Topo IV ATPase domain with an oligonucleotide[46]. Basic residues lining the transducer cavity and in particular residue R286 in the *E. coli* ATPase domain are thought to contribute to DNA capture and enzyme allostery[37]. The GHKL region of the ATPase domain shows a strong sequence identity across species due to the universal conservation of the catalytic motifs, with more variability across the transducer sequence (Supplementary Fig. 8). In *T. thermophilus*, the equivalent position K284 is pointing in the cavity and could potentially mediate an interaction with DNA; however, our experiments show that point mutations have no effect on the catalytic activities. In contrast, the analysis of the *E. coli* structure shows that R286 is engaged in an interaction network with E264 and R316 anchoring together secondary structures of the transducer and providing rigidity to the transducer terminal helix (Fig. 4b). In

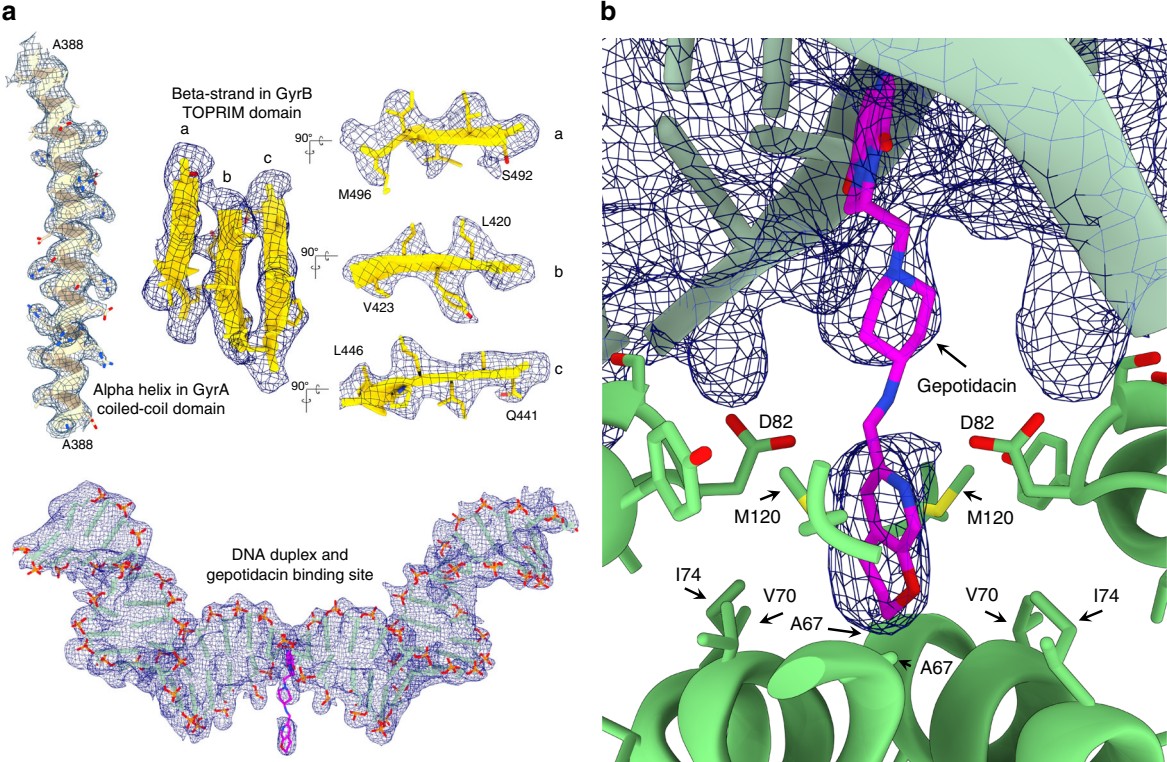

**Fig. 6** High-resolution features of the DNA-binding/cleavage domain bound to gepotidacin. **a** High-resolution features of the 4 Å cryo-EM map. An alpha helix and beta-sheet with well resolved side chains are shown on the upper panel. The 36 bp DNA duplex and gepotidacin are shown in the bottom panel. **b** Gepotidacin binding site. Electron density of the DNA and gepotidacin are shown in blue mesh. Residues in the direct vicinity of the compound are highlighted

*T. thermophilus*, the transducer is instead stabilized by a hydrophobic core where aliphatic side chains are replaced by aromatic residues and a different hydrogen bond network, notably involving H256 and Y287 (Fig. 4a). Rather than a direct effect involving DNA contact, we conclude that efficient allosteric transmission of ATP hydrolysis relies on the transducer rigidity that is maintained by species-specific interaction networks. It is not clear at this stage if DNA capture and transport is directly linked to the basic residues in the transducer and at which catalytic step molecular interaction with the T-segment intervenes. The full architecture of the complex in presence of a T-segment still needs to be determined to better understand the molecular determinants of DNA capture.

The complete DNA binding around the β-pinwheel surface generates a twisting of ~200° consistent with the reported wrapping around DNA gyrase pinwheel blades[47]. The structure of the β-pinwheel adopts a slight superhelical twist as previously observed in the crystal structure of the isolated domain[10] (Fig. 3b and Supplementary Fig. 6c). The highly conserved GyrA-box motif, QRRGGKG[48], located on the first blade of the β-pinwheels is essential for DNA wrapping (Fig. 3a, b). Complete deletion or alanine substitution of this motif completely abolishes the negative supercoiling activity[49]. By elucidating the spatial localization of the GyrA-box motif when DNA gyrase is bound to DNA, we show here why the GyrA-box motif is crucial in wrapping DNA. The position of the GyrA-box suggests that it acts as a clamp maintaining the DNA wrapped around the β-pinwheel with strong interaction that can compensate for the energetically unfavorable configuration of the twisted DNA (Fig. 3c). This explains the role of the GyrA-box behaving as a 'at the end of the wrap' mechanism, consistent with previous studies[50,51]. In addition, the analysis of the complex overall conformation shows that

the β-pinwheels are oriented in a way that the CTD tail could face the GyrB TOPRIM domain. The CTD sequence of *E. coli* DNA gyrase is composed of an unfolded and poorly conserved acidic tail starting at the end of the β-pinwheel structure (Supplementary Fig. 13a). Several biochemical studies have shown that the CTD tail stimulates DNA wrapping coupled with ATP hydrolysis[43,52]. Modeling of the 34 residues of the *E. coli* CTD tail shows that it could easily reach and contact the TOPRIM insertion domain (Supplementary Fig. 13b). Conformational changes of the DNA-gate from closed to pre-opening state may conduct a signal to the β-pinwheel through the tethered acidic tail promoting a β-pinwheel vertical movement. This interaction might also drive the relative orientation of the β-pinwheels wrapped with DNA, positioning the T-segment in the proper direction for transport to the DNA-gate.

The high-resolution features of the DNA-binding/cleavage domain structure now makes it possible to detect the density of small molecules and possibly catalytic ions in the cryo-EM map. We were able to observe the gepotidacin molecule inserted in the DNA on the two-fold axis at the GyrA dimer interface, as previously observed in crystal structures of the *S. aureus* DNA gyrase in complex with the NBTI[22,25,40,41]. Gepotidacin is currently in clinical trials for the treatment of *N. gonorrhoeae* infections[23,24], a species that is phylogenetically closer to *E. coli* and also harbors an isoleucine at this position in the gyrase sequence (Supplementary Fig. 11b, c). The comparison of the cryo-EM map with the electron density map of crystal structures[25] shows that the cryo-EM map information is of comparable quality in this region for the identification of small compounds (Supplementary Fig. 14).

Based on this study, we foresee that it is now possible to consider cryo-EM as a viable tool to perform structure-guided

drug design to target flexible complexes and identify new conformations of DNA gyrase that could so far not be obtained by conventional structural methods.

## Methods

**GyrB and GyrA expression and purification.** The sequence coding for the full-length E. coli GyrA (2-875) was inserted into a modified pET28b containing an N-terminal 10-His tag and a C-terminal Twin-strep tag. Overexpression was performed in E. coli BL21 (DE3) pRARE2 in LB medium containing 50 μg/mL kanamycin and 35 μg/mL chloramphenicol. Cells were induced with 0.35 mM IPTG after reaching an $OD_{600}$ 0.85 and protein was expressed at 37 °C for 4 h. Cells were harvested and resuspended in lysis buffer (20 mM Hepes, 500 mM NaCl, 20 mM imidazole, 10% v/v glycerol, pH 8.0) and lysed with three cycles of high-pressure disruption using EmulsiFlex-C3 at 1500 bars. The GyrA protein was purified by nickel-affinity chromatography on a manually packed XK 26/20 column (Pharmacia) with Chelating Sepharose 6 Fast Flow resin (GE Healthcare) bound to $Ni^{2+}$ ions. Elution was performed with the lysis buffer containing 250 mM imidazol pH 8.0 and eluted proteins were directly attached on a 10 ml Streptavidin Sepharose (GE Healthcare). Proteins were extensively washed with Strep buffer (20 mM Hepes, 60 mM NaCl, 1 mM EDTA, 1 mM DTT, 10% v/v glycerol, pH 8.0) and eluted with Strep buffer containing 3 mM Desthiobiotin (Sigma–Aldrich). Both 10-His tag and Twin-strep tag were subsequently cleaved by PreScission protease (P3C) and Tobacco Etch Virus (TEV) cleavage (mass ratio 1:1:50 P3C-TEV-GyrA) overnight at 4 °C. GyrA was then further purified by an anion exchange chromatography step using a HiTrap Q HP column (GE Healthcare). The protein was eluted with a linear gradient of 20 column volumes with buffer B (20 mM Hepes, 1 M NaCl, 1 mM EDTA, 1 mM DTT, 10% v/v glycerol, pH 8.0). Fractions containing GyrA were pooled and loaded on a Superdex S200 16/60 size-exclusion chromatography column (GE Healthcare) using 20 mM Hepes, 50 mM NaGlu, 50 mM KAc, 1 mM EDTA, 0.5 mM DTT, 10% v/v glycerol, pH 8.0. 37 mg of GyrA were obtained from 3 L of cultures. The GyrB (2-804) coding sequence was inserted into the same modified pET28b. Overexpression was performed in E. coli BL21 (DE3) pRARE2 in LB medium containing 50 μg/mL kanamycin and 35 μg/mL chloramphenicol. Cells were induced with 0.35 mM IPTG after reaching an $OD_{600}$ 0.85 and protein was expressed at 18 °C for 18 h. The GyrB purification procedure is as described above for GyrA. 10 mg of GyrB were obtained from 3 L of cultures.

**Full DNA gyrase reconstitution.** E. coli GyrA and GyrB were mixed at equimolar ratio to allow full DNA gyrase reconstitution. The complex was further purified on a Superdex S200 16/60 size-exclusion chromatography column (GE Healthcare) using cryo-EM buffer (20 mM Hepes, 30 mM NaGlu, 30 mM KAc, 2 mM MgAc, 0.5 mM TCEP, pH 8.0) (Fig. 1 and Supplementary Fig. 1).

**Nucleic acid preparation.** A double-nicked 180 bp DNA duplex was reconstituted using 2 phosphorylated asymmetric synthetic oligonucleotides[12] obtained from Sigma–Aldrich (Supplemental Table 1). Briefly, the nucleic acids were dissolved in DNAse-free water at 1 mM concentration. To assemble the double-stranded DNA, each oligo was mixed at 1:1 molar ratio, annealed by incubating at 95 °C for 2 min and then decreasing the temperature by 1 °C every 1 min until reaching 20 °C. The annealed doubly nicked DNA duplex was then buffer-exchanged in Hepes 20 mM pH 8.0 with a BioSpin 6 column (BioRad).

**Complex formation for cryo-EM.** The purified DNA gyrase was mixed with the 180 bp dsDNA at 1:1 molar ratio with a final protein and DNA concentration of 32 μM. The mixture was incubated for 10 min at 37 °C. Gepotidacin (GSK2140944, purchased at MedChemExpress) resuspended at 10 mM in 100% DMSO was added to reach a final concentration of 170 μM (1.7% DMSO). The mixture was incubated for 10 min at 37 °C. AMP-PNP (Sigma) was added to the complex at a final concentration of 340 μM. The fully reconstituted complex was further incubated 30 min at 30 °C. Finally, 8 mM CHAPSO (Sigma–Aldrich) was added to the complex. The sample was centrifuged 2 h at $16,000 \times g$ to remove potential aggregates.

**Cryo-EM grid preparation.** Quantifoil R-1.2/1.3 300 mesh copper grids were glow-charged for 20 s prior to the application of 4 μl of the complex. After 30 s of incubation, the grids were plunge-frozen in liquid ethane using a Vitrobot mark IV (FEI) with 95% chamber humidity at 10 °C.

**Electron microscopy.** Cryo-EM imaging was performed on a Titan Krios microscope operated at 300 kV (FEI) equipped with a K2 Summit direct electron camera (Gatan), a GIF Quantum energy filter (Gatan) operated in zero-energy-loss mode with a slit width of 20 $e^-$V, a Volta Phase Plate (FEI) and a CS corrector (FEI). Images were recorded in EFTEM nanoprobe mode with Serial EM[53] in super-resolution counting mode at nominal magnification of 130,000x with a super-resolution pixel size of 0.44 Å and a constant defocus target of −500 nm (Supplementary Fig. 2c). The VPP was advanced to a new position every 100 min (Supplementary Fig. 2b). Four

datasets were collected with a dose rate of 6 to 8 $e^-$/pixel/s (0.88 Å pixel size at the specimen) on the detector. Images were recorded with a total dose of 50 $e^-/Å^2$, exposure time between 7 to 10 s and 0.2 to 0.25 s subframes (35 to 50 total frames). A total of 11,833 movies were recorded after data collection of the 4 datasets.

**Data processing.** Data processing of each dataset was done separately following the same procedure until the 3D refinements when particles were merged. The super-resolution dose-fractionated subframes were gain-corrected with IMOD[54] and binned twice by Fourier-cropping, drift-corrected and dose-weighted using MotionCor2[55] yielding summed images with 0.88 Å pixel size. The contrast transfer function of the corrected micrographs was estimated using GCTF v1.06[56]. Thon rings were manually inspected for astigmatism and micrographs with measured resolutions worse than 4 Å were discarded yielding 8,701 remaining micrographs. Particles were automatically picked by reference-free Gaussian blob picking protocol in RELION2[29,30]. Taking together the 4 datasets, a total of 1,572,962 particles were selected. Four times binned particles from each dataset were separately subjected to two rounds of 2D classification in RELION2 to remove junk particles and contaminations resulting in a total of 479,667 particles for further processing. Particles from the four datasets were merged into one unique dataset followed by a re-extraction with centering in $3 \times 3$ binned format. A last round of 2D classification was then performed yielding a final particle stack of 338,616 particles. The subsequent particle stack was subjected to one round of 3D ab-initio classification in cryoSPARC[31]. After discarding the poor-quality models, the remaining ab-initio model resulted in a final dataset of 191,456 particles, with a class probability threshold of 0.9 (Supplementary Fig. 2a). The ab-initio model was low-pass filtered to 30 Å and was used as a reference for homogeneous refinement in cryoSPARC resulting in a 5.4 Å map. The refined particle coordinates were then used for local CTF estimation using GCTF v1.06 followed by re-extraction of $2 \times 2$ binned particles with centering. This new particle stack was subjected to a 3D auto-refinement in RELION2 using the ab-initio model low-pass filtered at 30 Å yielding a map with a global resolution of 5.0 Å (Supplementary Fig. 3). Local resolution showed a range of resolution from 4.0 Å in the DNA-binding/cleavage core to 9.0 Å in the ATPase domain and the GyrA β-pinwheel wrapping the DNA, indicative of high flexibility of these two modules. To obtain a final reconstruction of the full complex with well-defined densities for each domain, we performed 3D classification without alignment yielding several different classes. Particles from three classes were merged (94,633 particles). The subsequent $1 \times 1$-binned particles stack was refined in RELION2 without mask until late refinement iterations where a soft mask was applied to improve resolution. Post-processing of the map yielded a 6.6 Å resolution reconstruction of the overall complex (Supplementary Fig. 3).

A combination of different local approaches was used to identify different conformations and to improve local resolution of each domain. Since the ATPase domain was too small for a 3D focused refinement, we first performed a focused 3D classification of the ATPase domain with a soft mask and no alignment in RELION2. One class of 58,329 particles with a well-defined density of the ATPase domain was selected, followed by re-extraction of $1 \times 1$-binned particles yielding particles with pixel size of 0.88 Å/pixel. To facilitate an accurate alignment of the particles, this class was further refined in RELION2 using a soft mask around the ATPase domain and the DNA-binding/cleavage domain yielding a 5.9 Å global resolution map (ATPase-Core) (Supplementary Fig. 3).

Secondly, we performed a focused 3D classification of the GyrA β-pinwheel with a soft mask and no alignment in RELION2. One class of 45,040 particles with a well-defined density of the GyrA β-pinwheel was selected, followed by re-extraction of $1 \times 1$-binned particles yielding particles with pixel size of 0.88 Å/pixel. This class was further refined in RELION2 using a soft mask around the β-pinwheel and the DNA-binding/cleavage domain yielding a 6.3 Å resolution map (CTD-Core) (Supplementary Fig. 3).

Finally, a focused 3D auto-refinement was performed in RELION2 using a soft mask around the DNA-binding/cleavage domain. Post-processing of the map produced a 4.3 Å resolution reconstruction of the DNA-binding/cleavage domain. Then, a focused 3D classification of the DNA-binding/cleavage domain was performed. Two of the classes showed better angular accuracies and distinct closed (60,548 particles) and pre-opening conformations (53,655 particles) of the DNA-binding/cleavage domain. These two classes were further refined in RELION2 by focused 3D refinement with a C2 symmetry using $1 \times 1$ binned particles and gave reconstructions with global resolution of 4.0 and 4.6 Å after post-processing, respectively (Supplementary Fig. 3). The closed DNA-binding/cleavage domain was further refined by focused 3D refinement using a soft mask, which excludes the disordered TOPRIM insertion yielding a 4.0 Å map of high quality (Supplementary Fig. 3).

All reported resolutions are based on the gold standard FSC-0.143 criterion[57] and FSC-curves were corrected for the convolution effects of a soft mask using high-resolution noise-substitution[58] in RELION2 as well as in cryoSPARC (Supplementary Fig. 4a). All reconstructions were sharpened by applying a negative B-factor that was estimated using automated procedures[59]. Local resolution of maps was calculated using Blocres[60] (Supplementary Fig. 4c). The cryo-EM maps of the overall complex, ATPase-core, CTD-core and the DNA-binding/cleavage domain in closed (with and without TOPRIM insertion) and in pre-opening state have been deposited in the EM Data Bank under accession numbers EMD-4913, EMD-4914, EMD-4915, EMD-4910, EMD-4909, EMD-4912, respectively.

**Model building and refinement**. The EM reconstruction of the DNA-binding/cleavage domain lacking the TOPRIM insertion solved at 4.0 Å was of the best quality. Almost all side chains could be seen and Gepotidacin density was clearly visible. This map was used to refine a crystal structure of the DNA-binding/cleavage domain of the *E. coli* DNA gyrase[17]. A dimeric atomic model of the DNA-binding/cleavage domain was generated using PDB 3NUH in PyMol (Schrodinger L.L.C.). The subsequent atomic model was stripped of amino acids belonging to the TOPRIM insertion, all ions and water molecules, with all occupancies set to 1 and B-factors set to 50. First, the atomic model was rigid-body fitted in the filtered and sharpened map with Chimera[61]. A first round of real-space refinement in PHENIX[62] was performed using local real-space fitting and global gradient-driven minimization refinement. Then, 20 nucleic acids from the structure *of S. aureus* DNA gyrase[41] (PDB 5IWM) were copied and fitted into our atomic model. DNA sequence was modified according to the DNA used in our structure. Missing protein residues and nucleic acids were manually built in COOT[63]. Atomic model and constraint dictionary of gepotidacin (GSK-2140944) were generated with the Grade server (http://grade.globalphasing.org). Gepotidacin was then manually fitted in the empty electron density in COOT and then duplicated. Both duplicates were set to 0.5 occupancy. Several rounds of real-space refinement in PHENIX using restraints for secondary structure, rotamers, Ramachandran, and non-crystallographic symmetry were performed, always followed by manual inspection in COOT, until a converging model was obtained. Finally, B-factors were refined by a final round of real-space refinement in PHENIX using the same settings as before. After the refinement has converged, atomic coordinates of the TOPRIM insertion were added to the atomic model. It was further refined in the EM reconstructions containing the insertion density in closed and pre-open states using the same procedure. A half-map cross-validation was performed to define 1.5 as the best refinement weight in PHENIX allowing atom clash reduction and prevention of model overfitting (Supplementary Fig. 4e). All refinement steps were done using the resolution limit of the reconstructions according to the gold standard FSC-0.143 criterion[57]. Refinement parameters, model statistics and validation scores are summarized in Supplementary Table 2. The atomic models of the *E. coli* DNA-binding/cleavage domain in closed (with and without insertion) and pre-opening conformations have been deposited in the Protein Data Bank under accession numbers 6RKU, 6RKS, 6RKV, respectively.

For the overall complex, we used a combination of different EM reconstructions to build the DNA gyrase overall complex atomic model. The ATPase-Core structure solved at 5.9 Å was used to rigid-body fit in Chimera the ATPase domain crystal structure in complex with ADPNP[32] (PDB 1EI1) and the DNA-binding/cleavage domain refined in closed conformation. The quality of the electron density allowed to build in COOT the missing residues of the linker between the ATPase domain and the DNA-binding/cleavage domain. The CTD-Core structure solved at 6.3 Å was used to accurately rigid-body fit the β-pinwheel crystal structure[10] in Chimera. The 10 missing amino acids (564–574) following the GyrA-box were added using the modelling server Phyre2[64]. The quality of the electronic density allowed to build in COOT the missing nucleic acids around the β-pinwheel as well as the 10 amino acids linking the last GyrA residues to the β-pinwheel. Based on a secondary structure prediction, a part of the linker was built as an alpha helix (Fig. 3). The subsequent atomic model containing the ATPase domain, the DNA-binding/cleavage domain and one β-pinwheel was rigid-body fitted into the overall complex structure solved at 6.6 Å using Chimera. Then, a copy of the first β-pinwheel was fitted into the density of the second β-pinwheel in COOT. The missing nucleic acids around the second β-pinwheel as well as the missing 10 residues of the linker were manually built in COOT. The resulting atomic model was stripped of all ions and water molecules, with all occupancies set to 1 and B-factors set to 50. Finally, real-space refinement of the atomic model against the overall complex structure was performed in PHENIX using rigid-body and global gradient-driven minimization refinement. Resolution limit for refinement was set according to the gold standard FSC-0.143 criterion[57]. Half-map cross-validations were also performed (Supplementary Fig. 4e). Refinement parameters, model statistics and validation scores are summarized in Supplementary Table 2. The atomic model of the overall structure has been deposited in the Protein Data Bank under accession numbers 6RKW. All the Figures were created with Chimera[61], ChimeraX[65] and PyMol (Schrodinger L.L.C.).

**E. coli GyrB R286K, R286Q, and E264A purification**. The modified pET28b used for wild-type *E. coli* GyrB overexpression was mutated by site-directed mutagenesis using the QuikChange XL Site-Directed Mutagenesis kit (Agilent) in order to generate three plasmids harboring R286K, R286Q, or E264A mutations (Supplementary Table 4). Overexpression and purification procedures for the three mutants are identical to the wild-type GyrB described above in the Methods section.

**T. thermophilus GyrB K284R, K284Q purification**. The modified pET28b used for wild-type *T. thermophilus* GyrB overexpression[12] was mutated by site-directed mutagenesis using the QuikChange XL Site-Directed Mutagenesis kit (Agilent) in order to generate two plasmids harboring K284R or K284Q mutations (Supplementary Table 4). Overexpression and purification procedures for the two mutants are identical to the *E. coli* wild-type GyrB described above in the Methods section.

**DNA supercoiling assay**. An increasing concentration of DNA gyrase (GyrA$_2$B$_2$) was incubated at 37 °C with 6 nM of relaxed pUC19 plasmid in a reaction mixture containing 20 mM Tris-acetate pH 7.9, 100 mM potassium acetate, 10 mM magnesium acetate, 1 mM DTT, 1 mM ATP, 100 µg/ml BSA. After 30 min, reactions were stopped by addition of SDS 1%. Agarose gel electrophoresis was used to monitor the conversion of relaxed pUC19 to the supercoiled form. Samples were run on a 0.8% agarose, 1X Tris Borate EDTA buffer (TBE) gel, at 6 V/cm for 180 min at room temperature. Agarose gels were stained with 0.5 mg/ml ethidium bromide in 1X TBE for 15 min, followed by 5 min destaining in water. DNA topoisomers were revealed using a Typhoon (GE Healthcare).

**ATPase activity assay**. ATP hydrolysis is measured by following the oxidation of NADH mediated by pyruvate kinase (PK) and lactate dehydrogenase (LDH). The absorbance was monitored at 340 nm over 600 s at 37 °C with a Shimadzu 1700 spectrophotometer. Reactions were recorded in triplicates with 50–100 nM of GyrA$_2$B$_2$ and 16 nM linear DNA (pCR-blunt) in 500 µl of a buffer containing 50 mM Tris HCl pH 7.5, 150 mM potassium acetate, 8 mM magnesium acetate, 7 mM BME, 100 µg/mg of BSA, 4U/5U of PK/LDH mixture, 2 mM PEP, and 0.2 mM NADH.

**Multiple alignment and evolutionary conservation of residues**. GyrB ATPase/Transducer protein sequences from 30 species (Uniprot codes: GYRB_STRCO, *Streptomyces coelicolor*; A0A3A2T2C3_LISMN, *Listeria monocytogenes*; W8F4Q2_AGRT4, *Agrobacterium tumefaciens*; Q83FD5_COXBU, Coxiella burnetii; GYRB_ECOLI, *Escherichia coli*; GYRB_NEIGO, *Neisseria gonorrhoeae*; GYRB_SALTY, *Salmonella typhimurium*; Q7V9N3_PROMA, *Prochlorococcus marinus*; GYRB_STRR6, *Streptococcus pneumoniae*; GYRB_MYCPN, *Mycoplasma pneumoniae*; GYRB_SHIFL, *Shigella flexneri*; GYRB_PSEAE, *Pseudomonas aeruginosa*, Q18C89_PEPD6, *Clostridium difficile*; GYRB_RICFE, *Rickettsia felis*; Q8CZG1_YERPE, *Yersinia pestis*; GYRB_BORBU; *Borrelia burgdorferi*; GYRB_ENTFA, *Enterococcus faecalis*; Q8YEQ5_BRUME, *Brucella melitensis*; GYRB_STAAU, *Staphylococcus aureus*; Q7VQU8_BLOFL, *Blochmannia floridanus*; GYRB_HAEIN, *Haemophilus influenzae*; GYRB_MYCTU, *Mycobacterium tuberculosis*, GYRB_THEMA, *Thermotoga maritima*; GYRB_THET8, *Thermus thermophilus*; GYRB_BACSU, *Bacillus subtilis*; GYRB_HELPY, *Helicobacter pylori*; GYRB_CHLTR, *Chlamydia trachomatis*; GYRB_VIBCH, *Vibrio cholerae*; GYRB_THEAQ, *Thermus aquaticus*; GYRB_AQUAE, *Aquifex aeolicus*) were aligned using the Clustal Omega server (EMBL-EBI). The subsequent alignment was used to plot the amino acids evolutionary conservation on the ATPase/transducer structure (PDB ID 1EI1) using the ConSurf server (http://consurf.tau.ac.il). The phylogenetic tree was generated by neighbour-joining method using the multiple alignment in Clustal Omega.

**Reporting summary**. Further information on research design is available in the Nature Research Reporting Summary linked to this article.

## Data availability
Model coordinates and density maps are available in the Protein Data Bank (PDB ID 6RKS [https://www.rcsb.org/structure/6RKS], 6RKU [https://www.rcsb.org/structure/6RKU], 6RKV [https://www.rcsb.org/structure/6RKV], 6RKW [https://www.rcsb.org/structure/6RKW]) and the EM Data Bank (EMD-4909 [https://www.ebi.ac.uk/pdbe/entry/emdb-EMD-4909], EMD-4910 [https://www.ebi.ac.uk/pdbe/entry/emdb/EMD-4910], EMD-4912 [https://www.ebi.ac.uk/pdbe/entry/emdb/EMD-4912], EMD-4913 [https://www.ebi.ac.uk/pdbe/entry/emdb/EMD-4913], EMD-4914 [https://www.ebi.ac.uk/pdbe/entry/emdb/EMD-4914], EMD-4915 [https://www.ebi.ac.uk/pdbe/entry/emdb/EMD-4915]). The source data underlying Figs. 1a, 4c, d and Supplementary Fig. 9b are provided as a Source Data file. Other data are available from the corresponding author upon reasonable request.

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

## Acknowledgements

We thank the teams of B. Klaholz and P. Schultz, as well as A. Weixlbaumer and X. Guo for useful technical suggestions and help with computational resources. This work was supported by the Fondation ARC and the grant ANR-10-LABX-0030-INRT (managed by the Agence Nationale de la Recherche under the frame programme Investissements d'Avenir ANR-10-IDEX-0002-02). The authors acknowledge the support and the use of resources of the French Infrastructure for Integrated Structural Biology FRISBI ANR-10-INBS-05 and of Instruct-ERIC. Computational resources were provided by the Méso-centre de Calcul (University of Strasbourg).

## Author contributions

A.V.B. and V.L. conceived the study and designed the experiments; A.V.B. and C.L. performed the experiments; A.V.B. and J.O. performed the cryo-EM data collection; A.V.B. processed the cryo-EM data and built the atomic models; A.V.B. and V.L. analyzed and interpreted the data; A.V.B. and V.L. wrote the manuscript.

## Competing interests

The authors declare no competing interests.
