## [Peer Review File · Nature Communications]

Reviewers' comments:

Reviewer #1 (Remarks to the Author):

The manuscript by Lamour et al. describes the cryo-EM structure of the complete *E. coli* gyrase complex with DNA. The work is timely and addresses important issues in the field. The authors provide structural evidence for the mechanism by which the GyrA-box motif exerts its actions in binding and bending DNA. It also provides information regarding the overall catalytic activity of gyrase. Finally, the work shows that interactions of the enzyme with gepotidacin [a Novel Bacterial Gyrase Inhibitor (NBTI) that is currently in clinical trials] as determined by cryo-EM, are consistent with a recent crystal structure. Overall, the work by the Lamour laboratory is well crafted, well written, and makes an important contribution to the topoisomerase field. The manuscript should definitely be of interest to readers of Nature Communications. I have only minor issues. Specific comments follow

1. In Figure S9, the authors indicate that the k_{cat} of GyrB R286K is the same as that of the wild-type enzyme. However, in Fig. 5b, the mutant enzyme appears to supercoil DNA 3-5 times faster than does wild-type gyrase. The authors need to resolve this discrepancy.
2. Especially in light of the fact that gepotidacin is currently in clinical trials, the authors need to describe the source of the compound that they utilized in their paper.
3. Earlier studies have demonstrated that human topoisomerase II α preferentially relaxes positively supercoiled DNA over negatively supercoiled molecules and that the ability to do so lies in the C-terminal domain in the vicinity of the first blade [McClendon et al. (2008) *Biochemistry* 47, 13169]. Does the present structural work on the role of GyrA-box offer any insight into the function of the human enzyme?

Reviewer #2 (Remarks to the Author):

The manuscript of Vanden Broeck et al. presents the structure of *E. coli* DNA gyrase. The authors succeeded in determining the complete structure in complex with DNA and an antibiotic using cryo-electron microscopy. Overall, this is a very important paper. The structure is interesting for several reasons. It is only the second structure of a complete type II topoisomerase (the first one, at lower resolution and from another species, e.g. *T. thermophilus*, was determined by the same group). It also represents a step towards a better understanding of the conformational changes required for cutting and resealing DNA by bacterial topoisomerases. Interestingly, the proposed mechanisms can be extrapolated to all type IIA topoisomerases.

I would recommend publication in Nature Communications, provided that a few issues are addressed.

1. There is a myriad of typo mistakes, that render the manuscript difficult to read. Hereafter are only some examples:
 - Gyrase is a common noun, please avoid using capital letter, the same accounts also for gepotidacin or aminoacids
 - Please remove the capital letter from "Coli" in the movie (at the end of the movie)
 - P2. discreet conformation?
 - An infinite number of hyphens are missing all along the manuscript (GyrA-box, ATP-binding, DNA-gate, DNA-binding/cleavage domain, ...)
2. The penultimate paragraph of the introduction is confusing. The authors introduce the fact that the *E. coli* gyrase possesses a specific 170-aa insertion (for this point, see also "minor remarks") and then switch to the introduction of novel bacterial topoisomerase inhibitors, which, indeed, are not specific to *E. coli*. The aim of this paragraph should be clarified.
3. The orientation of the pinwheel (CTD of GyrA) with respect to the core of the enzyme is an

important and crucial discussion, notably because the pinwheels are responsible for DNA wrapping during the enzymatic cycle. Even if the arguments seem to be incontestable, I am not totally convinced that it is not impossible to orientate it upside down (rotation of 180 degrees), the N-terminal linker being flexible enough to accommodate this reverse conformation. Even in the movie, one would be able to imagine the reverse orientation. Perhaps the authors could provide more persuasive figure(s). Regarding the structure of the CTD, does it adopt a spiral shaped pinwheel? In the figures, the CTD seem to be quite flat. Whether the spiral or flat shapes correspond to a given conformation or the inherent structure is still an open question. As the CTD plays an important role in the conformational movements they could exemplify, it would be interesting to discuss this point. Three additional remarks concerning the CTD. (i) What is the color code of the CTD? It does not seem to follow something rationale. (ii) The authors should define or remind what is the top and the bottom of the pinwheels. (iii) DNA seems to lie around the CTD without interaction. The authors should comment this point.

4. Concerning connections between the domains, the authors address this point starting with "The connection between the ATPase domain and DNA binding cleavage domain..." (P9) which is confusing. Two aspects seem to be combined, localization of the ATPase domain (the ATPase domain is sitting on the catalytic core) and the "physical link" (the loop) of the ATPase domain to the TOPRIM domain (that was correctly introduced in the previous section). This should be clarified, by notably modifying the inset in Fig. 2a, so as to show the two ends of the linker. The linker between the ATPase and the TOPRIM domains was clearly visible in a crystal structure of *M. tuberculosis* DNA gyrase. Are they very different?

5. Concerning the explanation as to why the *E. coli* mutant R286Q has a reduced negative supercoiling activity, the argument is indirect, using the fact that the *Thermus* transducer is stabilized by a hydrophobic core involving 5 residues. To me, this argument sounds not totally convincing. These 5 residues are also hydrophobic in *E. coli* (L, W, F, M and A). The equivalent position in *T. thermophilus* is Ala264 (not stated in the text), corresponding to a loss of the salt bridge. Could it be possible to analyze the mutant *Ec* E264A, in order to mimic the *T. thermophilus* enzyme at this position and to confirm this hypothesis? *T. thermophilus* is the only sequence in the alignment that possesses an alanine a position 264. *Thermus aquaticus* possesses a threonine. Is *T. thermophilus* an exception? What about other *Deinococcus-Thermus* bacteria?

I would also simplify the paragraph that discusses this point in the discussion section. The discussion concerning the lysine/arginine replacement is not really useful with respect to the importance of the loss of the salt bridge.

One additional remark, the numbering of these residues seems mixed up. Why Lys284 in *T. thermophilus* and not Lys286? Based on the alignment, there is no insertion/deletion between *E. coli* and *T. thermophilus*.

6. In the discussion section, I do not totally agree with the statement "For decades, the opening of the DNA-gate was predicted to follow a "book-opening" motion, in which the two halves of the DNA-gate simply remain along the axis of G-segment 2,3,41. However, it has been shown recently that the DNA-gate opening mechanism is more subtle and is achieved by a "sliding and swiveling" motion". Even if the animation presented in the movie published by Berger in 1996 shows a simple "book-opening" motion, the structure of the breakage-reunion domain of the *S. cerevisiae* topoisomerase II (PDB code 1BGW, Berger et al., 1996) clearly indicates that the motion is closer to a screwing motion following an elliptic path, similar to the sliding and swiveling motion. We could confirm this by calculations of the transition path we performed using the Path Optimization and Exploration approach (in collaboration with Arnaud Blondel, Institut Pasteur, unpublished results). I would recommend reviewing this section.

7. Furthering the topic of the conformational changes of the DNA-binding and cleavage domain during opening of the DNA-gate, the authors present two reconstruction corresponding to two states, a closed and a pre-opening state. In Fig. 6a, it is difficult to apprehend the pre-opening state. Was the superimposition performed on the grey monomers? Fig. 6b should help to better clarify this point, but the structural differences in this part of the figure are also difficult to see.

In addition, we cannot neglect the fact that, if one takes into account all the structural data concerning the DNA-binding and cleavage domain of type II topoisomerases, including structures of

gyrases, topo IV and topo II, more than 50 structures are available. Even if it would be a huge work to analyse all these structures, did the authors check if there is a structure that could be assimilated to a pre-opening conformation?

8. When comparing to the *T. thermophilus* cryo-EM structure, the authors propose that the *E. coli* conformation precedes the conformation of the *T. thermophilus* one. This is highly hypothetical. A bending angle of the segment of 150° should be energetically unfavorable. As a side remark, how are the -10 and 10 degrees angle measured?

In addition, the sentence « Interestingly, a recent single-molecule study described a Ω state, possibly similar to the *E. coli* complex and occurring before the α state in the catalytic cycle, more reminiscent of the *T. thermophilus* conformation” is difficult to follow. I would recommend clarifying this sentence. They also mentioned the acidic extreme-CTD tail of the *E. coli* enzyme. Is it visible in the density? What about the CTD tail of *T. thermophilus*?

10. The figures could be slightly improved.

- The inset of figure 2 has already been mentioned
 - Color code of the CTD in figure 3 has already been mentioned
 - In figure 4, the labeling should be improved (placement of the value angle labels, space before Angström, ...)
 - The shading in figure 5d is missing, rendering it flat
 - The clarity of figure 6 has already been mentioned
 - In Figure S8, the position in yellow is totally impossible to read
11. Some titles are misleading or inaccurate, especially the one of the last “Results” section that deals with drug and magnesium binding. I would recommend going through all the titles.

Minor remarks:

- P3. ADPNP is not the only non-hydrolysable analog of ATP
- P4. “Defined alpha helices”: which ones?
- P5. “GyrB TOPRIM insertion relative to the catalytic domains”. The catalytic domain is composed of the TOPRIM domain and the N-terminal domain of GyrA (which has several names in the literature), I recommend to reformulate this sentence (see also last point of the minor remarks)
- P6. “the atomics models of the GyrA and GyrB domains” Precise *E. coli*
- ATPase head: whole or part of the ATPase domain? Please, clarify
- Legend of figure 3: a space is missing (130bp)
- With the growing number of sequenced genomes, it would be very interesting to mention to what extent this insertion exists in bacterial gyrases. Is it found in all gyrases of the *Escherichia* genus? The Enterobacteriaceae family? The Enterobacterales order?
- What is included in the term “DNA-binding/cleavage domain”? The N-terminal domain of GyrA? The N-terminal domain of GyrA and the TOPRIM domain”. The literature is highly inconsistent with respect to this terminology. It would be worse to clarify it.

Signed,
Claudine Mayer

Reviewer #3 (Remarks to the Author):

Broeck et al. present structure of a complete Gyrase-DNA complex from *E. coli* using phase-plate single particle cryo-EM, revealing the overall architecture of this extremely flexible and essential DNA topoisomerase at great details for the first time. This is beautiful work and a big advance in the field. In particular, the galleries of structures generated from this study revealed an allosteric movement of the complex by comparing the closed versus the pre-opening states. Furthermore, the authors were able to identify the antibiotic location, thanks to their near atomic resolution density map produced from state-of-the-art data collection and analysis. The manuscript is well written and they have done a

good job selecting figures that illustrate important features of the structure models. All details leading to the cryo-EM reconstructions, models and fittings are very clearly described. There is simply nothing to be complained about the technical approaches chosen by the authors. The results reported here will guide the next few years of advances in the study of DNA topoisomerase; once published this work seems destined to become a classic.

I have only a minor comment that might guide improvement during preparation of a final version of the manuscript. In figure 4, the authors attempted to compare the architecture of DNA Gyrase complex and the associated DNA topology from *E. coli* (this study) and *T. Thermophilus*. However, the cryo-EM structure of the *Thermophilus* Gyrase was determined prior to the resolution revolution and it was limited to 23 Å-16.9 Å at the time. At this resolution, it's going to be hard to confidently delineate the path of the DNA. Indeed, in the rigid body docking, there was clear empty density in the *Thermophilus* structure which could fit the G-segment in a similar way as in the *E. coli* structure. Unless the authors can present higher resolution maps for the *Thermophilus* Gyrase structure, it would be wise to not put such an emphasis on the distinct mode of DNA engagement and bending between the two similar systems. In summary, this is an impressive piece of work.

Title: "Cryo-EM structure of the complete *E. coli* DNA Gyrase nucleoprotein complex"

Author(s): Vanden Broeck Arnaud, Lotz Christophe[§], Ortiz Julio, Lamour Valerie*

[§] added during reviewing

* Corresponding author

Point by point answer to reviewers' comments

We thank all the reviewers for the time they have taken to carefully review our work and for their comments, which helped us to improve our manuscript.

Reviewers' comments:

Reviewer #1 (Remarks to the Author):

The manuscript by Lamour *et al.* describes the cryo-EM structure of the complete *E. coli* gyrase complex with DNA. The work is timely and addresses important issues in the field. The authors provide structural evidence for the mechanism by which the GyrA-box motif exerts its actions in binding and bending DNA. It also provides information regarding the overall catalytic activity of gyrase. Finally, the work shows that interactions of the enzyme with gepotidacin [a Novel Bacterial Gyrase Inhibitor (NBTI) that is currently in clinical trials] as determined by cryo-EM, are consistent with a recent crystal structure. Overall, the work by the Lamour laboratory is well crafted, well written, and makes an important contribution to the topoisomerase field. The manuscript should definitely be of interest to readers of Nature Communications. I have only minor issues. Specific comments follow.

We thank the reviewer for the positive appreciation of our work and for the time taken to review our manuscript. We have answered below the remarks and comments.

1. In Figure S9, the authors indicate that the k_{cat} of GyrB R286K is the same as that of the wild-type enzyme. However, in Fig. 5b, the mutant enzyme appears to supercoil DNA 3-5 times faster than does wild-type gyrase. The authors need to resolve this discrepancy.

We agree with the reviewer that this point needs to be clarified. The GyrA₂B₂ GyrB R286K exhibits a slightly higher ATPase activity (1.25-fold) compared to the WT (K_{cat} WT: 6.8 s⁻¹; R286K 8.4 s⁻¹) (Fig. S9). In the negative supercoiling assay (Fig. 5), the GyrA₂B₂ GyrB R286K also shows a slightly higher activity compared to the WT. The GyrB R286K and WT enzymes completely form negative supercoiled plasmids at 0.5 nM and 1.25 nM, respectively. The increased negative supercoiling activity of the GyrB R286K is rather in the 2-2.5-fold range than the 3-5-fold mentioned by the

reviewer. This non-linear correlation between the ATPase activity and the supercoiling activity is a common phenomenon observed among allosteric enzymes. As shown by Gross *et al.* in 2003¹, GyrB with P79A and K103A substitutions (residues positioned in the direct vicinity of the ATP binding site) retained significant levels of ATPase activity yet demonstrated no DNA supercoiling activity. It is not to exclude that the opposite is also true. A small increase in the ATPase activity can have a more pronounced effect on the negative supercoiling activity due to the perturbation of hydrogen bond networks affecting allostery.

Overall, the mutation _{GyrB}R286K seems to increase the ATPase (1.25-fold) and negative supercoiling (2.5-fold) activities of the enzyme. In this regard, we primarily wrote the following sentence in the manuscript: “The *Ec* R286K mutant **is also not impaired** in any of the tested activities compared to the WT suggesting either side chain is compatible with the catalytic activities in *E. coli*”.

To clarify this point, we now have rephrased this sentence: “The *Ec* R286K mutant **exhibits slightly higher ATPase and DNA supercoiling activities** compared to the WT suggesting either side chain is compatible with the catalytic activities in *E. coli*.”

1. Gross CH, Parsons JD, Grossman TH, Charifson PS, Bellon S, Jernee J, Dwyer M, Chambers SP, Markland W, Botfield M, Raybuck SA. (2003) Active-site residues of Escherichia coli DNA gyrase required in coupling ATP hydrolysis to DNA supercoiling and amino acid substitutions leading to novobiocin resistance. *Antimicrob Agents Chemother.* 2003 Mar;47(3):1037-46.

2. Especially in light of the fact that gepotidacin is currently in clinical trials, the authors need to describe the source of the compound that they utilized in their paper.

The gepotidacin (GSK2140944) was purchased at MedChemExpress (Cat. No.: HY-16742). We added this information in the method section: “Gepotidacin (GSK2140944, purchased at MedChemExpress) resuspended at 10 mM in 100% DMSO was added to reach a final concentration of 170 μ M (1.7% DMSO)”.

3. Earlier studies have demonstrated that human topoisomerase II alpha preferentially relaxes positively supercoiled DNA over negatively supercoiled molecules and that the ability to do so lies in the C-terminal domain in the vicinity of the first blade [McClendon et al. (2008) *Biochemistry* 47, 13169]. Does the present structural work on the role of GyrA-box offer any insight into the function of the human enzyme?

Both DNA gyrase and human topoisomerase II alpha have a preference for positively supercoiled DNA as a starting substrate. DNA gyrase can relax positively supercoiled DNA in an ATP-independent manner faster than it can introduce negative supercoils into DNA². The C-terminal domains (CTD) of DNA gyrase are composed of 6 blades. The first blade comprises the GyrA-box, a conserved consensus sequence (QRRGGKG) composed of positively charged amino acid residues. The 5 other blades possess degenerated GyrA-boxes, also containing positively

charged amino acid residues, which are less conserved. We showed in our structure that the GyrA-box is positioned at the end of the DNA path around the CTD. Its unique localization is responsible for maintaining the DNA curvature resulting in the formation of a positive supercoiled crossover of the T-segment above the G-segment before the gyrase engages in negative supercoiling.

As mentioned by the reviewer, human topoisomerase II alpha preferentially relaxes positively supercoiled DNA over negatively supercoiled molecules³. This selectivity for positive supercoiled DNA lies into clusters of positively charged amino acid residues present in the CTD. Deletion of these clusters abolishes the preference of the enzyme for positively supercoiled DNA². It cannot be excluded that the clusters of positively charged amino acid residues present in the CTD of the human topoisomerase II alpha could have the same function as of the GyrA-boxes of DNA gyrase and play a role in sensing positive supercoiled substrate for relaxation. However, the amino acid sequences of the CTD of the bacterial DNA gyrase and the human topoisomerase II alpha are not evolutionary conserved. Besides, the CTD of the DNA gyrase adopts a 3-dimensional structure, while the structure predictions for the CTD of the human topoisomerase II suggest that this domain is unfolded and was shown to be the site of multiple post-translational modifications⁴.

Without a complete structure of the human topoisomerase II alpha, including the CTDs and a DNA substrate, it remains difficult to understand the structural function of the eukaryotic CTD and how it plays a role in DNA substrate topology recognition.

2. Ashley RE, Dittmore A, McPherson SA, Turnbough CL Jr, Neuman KC, Osheroff N. (2017). Activities of gyrase and topoisomerase IV on positively supercoiled DNA. *Nucleic Acids Res.* 2017 Sep 19;45(16):9611-9624.
3. McClendon AK, Gentry AC, Dickey JS, Brinch M, Bendtsen S, Andersen AH, Osheroff N. (2008) Bimodal recognition of DNA geometry by human topoisomerase II alpha: preferential relaxation of positively supercoiled DNA requires elements in the C-terminal domain. *Biochemistry.* 2008 Dec 16;47(50):13169-78
4. Bedez C*, Lotz C*, Batisse C, Vanden Broeck A, Stote RH, Howard E, Pradeau-Aubretton K, Ruff M, Lamour V. (2018) Post-translational modifications in DNA topoisomerase 2α highlight the role of a eukaryote-specific residue in the ATPase domain. *Sci Rep.* 8(1):9272.

Reviewer #2 (Remarks to the Author):

The manuscript of Vanden Broeck *et al.* presents the structure of *E. coli* DNA gyrase. The authors succeeded in determining the complete structure in complex with DNA and an antibiotic using cryo-electron microscopy. Overall, this is a very important paper. The structure is interesting for several reasons. It is only the second structure of a complete type II topoisomerase (the first one, at lower resolution and from another species, e.g. *T. thermophilus*, was determined by the same group). It also represents a step towards a better understanding of the conformational changes required for cutting and resealing DNA by bacterial topoisomerases. Interestingly, the proposed mechanisms can be extrapolated to all type IIA topoisomerases. I would recommend publication in Nature Communications, provided that a few issues are addressed.

We thank the reviewer for taking the time to carefully review our manuscript. We have responded below to all the remarks and comments:

1. There is a myriad of typo mistakes, that render the manuscript difficult to read. Hereafter are only some examples:

- Gyrase is a common noun, please avoid using capital letter, the same accounts also for gepotidacin or aminoacids.

These discrepancies have been addressed throughout the manuscript and in the supporting information.

- Please remove the capital letter from “Coli” in the movie (at the end of the movie).

This mistake has been corrected and a new version of the Supplementary Movie 1 has been provided.

- P2. discreet conformation?

By “discreet conformations”, we meant “slightly different, but detectable, intermediate conformations”. To our knowledge, it is a common term used to emphasize the oscillation between different states of a protein, which is the case with allosteric complexes. Data processing of cryo-electron microscopy images allows now to distinguish between multiple conformations of a flexible complex if the resolution is high enough. It was not the case with the *T. thermophilus* DNA gyrase⁵ but it is now with the *E. coli* DNA gyrase (this manuscript).

5. Papillon J, Ménétret JF, Batisse C, Hélye R, Schultz P, Potier N, Lamour V. (2013) Structural insight into negative DNA supercoiling by DNA gyrase, a bacterial type 2A DNA topoisomerase. *Nucleic Acids Res.* 41(16):7815-27.

- An infinite number of hyphens are missing all along the manuscript (GyrA-box, ATP-binding, DNA-gate, DNA-binding/cleavage domain, ...)

We thank the reviewer for bringing these discrepancies to our attention. We have now corrected these typos and used the same terms throughout the whole manuscript and in the supporting information.

2. The penultimate paragraph of the introduction is confusing. The authors introduce the fact that the *E. coli* gyrase possesses a specific 170-aa insertion (for this point, see also “minor remarks”) and then switch to the introduction of novel bacterial topoisomerase inhibitors, which, indeed, are not specific to *E. coli*. The aim of this paragraph should be clarified.

We agree with the reviewer that there is indeed no direct connection between the 2 notions. To clarify this part in the introduction, we separated the sentences describing the inhibitors targeting the DNA gyrase in a new paragraph, apart from the one describing the TOPRIM insertion. This should improve the readability of the introduction.

3. The orientation of the pinwheel (CTD of GyrA) with respect to the core of the enzyme is an important and crucial discussion, notably because the pinwheels are responsible for DNA wrapping during the enzymatic cycle. Even if the arguments seem to be incontestable, I am not totally convinced that it is not impossible to orientate it upside down (rotation of 180 degrees), the N-terminal linker being flexible enough to accommodate this reverse conformation. Even in the movie, one would be able to imagine the reverse orientation. Perhaps the authors could provide more persuasive figure(s). Regarding the structure of the CTD, does it adopt a spiral shaped pinwheel? In the figures, the CTD seem to be quite flat. Whether the spiral or flat shapes correspond to a given conformation or the inherent structure is still an open question. As the CTD plays an important role in the conformational movements they could exemplify, it would be interesting to discuss this point.

The densities for the pinwheel itself and of the connection between the pinwheel and DNA-binding/cleavage domain are unambiguous in the EM density map. It is worth noting that the cryo-EM maps are unbiased as they are obtained without any prior model, they result from a completely ab-initio 3D reconstruction. It is indeed not plausible that the pinwheel would be rotated by 180° since no density is visible **on the opposite face of the pinwheel** where the linker between the pinwheel and DNA-binding/cleavage domain would then connect. In addition, the number of amino-acids in the linker is compatible with the distances separating the DNA-binding/cleavage domain from the pinwheel. It would not be able to stretch further to reach the top of the pinwheel in a 180° orientation. Alternatively, if the pinwheel would rotate on itself, presuming the linker would stay in its actual density, this would generate a steric clash between the linker and the blades and would be inconsistent with the observed densities. Fig. 2a, 2b and 3c clearly showed

how the pinwheel is connected. For the sake of clarity, we provided another zoom on this area in a new panel Fig. 2e.

We confirm that the pinwheel adopts a slight superhelical twist in the context of the full-length *E. coli* gyrase DNA-bound complex as observed by Ruthenburg AJ and colleagues⁶ on the isolated pinwheel (PDB ID 1zi0). In this area of the EM map, the local resolution is slightly lower due to the flexibility of the pinwheels, as a consequence only rigid body refinement was applied. To better display this feature, we have added a new panel in Fig. 3b with a close-up on the pinwheel.

To illustrate the comparison between the CTD structure alone and within the full length complex, we now provide a superimposition as a supplementary figure (supplementary Fig. 7c) and added a comment in the discussion: “The structure of the β -pinwheel adopts a slight superhelical twist as previously observed in the crystal structure of the isolated domain¹⁰.”

10. Ruthenburg AJ, Graybosch DM, Huetsch JC, Verdine GL. (2005) A superhelical spiral in the *Escherichia coli* DNA gyrase A C-terminal domain imparts unidirectional supercoiling bias. *J Biol Chem* 280, 26177-84

Three additional remarks concerning the CTD.

(i) What is the color code of the CTD? It does not seem to follow something rationale.

The color code used for the CTD in Fig 3 is a rainbow coloring, starting with the N-terminal part in blue and ending with the C-terminal part in red. This color code allows the direct visualization of the N-terminal linker and the C-terminal end, as well as the discrimination of the different blades.

We also agree that the coloring of the schematic representation of former Fig. 3b might have been confusing when comparing with the structure representation in Fig. 3a. We initially thought it would be useful to have a schematic representation to show the connections between the blades as it is usually represented in the field. Since many publications have detailed this structure element for various organisms, we decided to remove this representation from panel 3b to avoid any confusion. To clarify Fig. 3a, we have written the blade numbers with the same color as the corresponding blade.

(ii) The authors should define or remind what is the top and the bottom of the pinwheels.

We agree this was not clearly stated in the results section describing the positioning of this domain in the EM map. We have reformulated this part in the main text: “We could also build the linker between the C-terminal end of the GyrA coiled-coil domains and the β -pinwheels on each side (Fig. 2e and Supplementary Fig. 5). The quality of the density allows to distinguish the upper from the lower face of the pinwheel disk where the N- and C-terminal ends are located. The tracing

of the linker connecting to the N-terminal end of the pinwheel in blade 1 enabled the precise orientation of the crystal structure of the *E. coli* β -pinwheel in the EM map ¹⁰ (Fig. 2e and Fig. 3).” We now also provide a close-up on the pinwheel density in the new panel Fig. 2e.

(iii) DNA seems to lie around the CTD without interaction. The authors should comment this point.

Fig. 3d may be misleading due to the perspective of this view. This figure is only meant to show the respective orientation of the 2 pinwheels with regards to the DNA wrapping. The EM map around the pinwheel is well connected to the density of the DNA as seen in Fig. 2a (now renumbered Fig. 2b), Fig. 3b (now renumbered Fig. 3b) or supplementary Fig. 5c. The structure is complementary to the DNA curvature which density wraps around the pinwheel. It's only close to the DNA extremities that the DNA does not completely interact with the pinwheel surface since the DNA fragment is linear and its extremities are disordered. To better illustrate the fit between the pinwheel structure and the DNA curvature, we have zoomed on the pinwheel density on a new panel 3b and represented the pinwheel structure in a surface representation in panel 3b and revised panel 3c.

4. Concerning connections between the domains, the authors address this point starting with “The connection between the ATPase domain and DNA binding cleavage domain...” (P9) which is confusing. Two aspects seem to be combined, localization of the ATPase domain (the ATPase domain is sitting on the catalytic core) and the “physical link” (the loop) of the ATPase domain to the TOPRIM domain (that was correctly introduced in the previous section). This should be clarified, by notably modifying the inset in Fig. 2a, so as to show the two ends of the linker.

We have transferred the inset of Fig. 2a into a new panel 2d, zooming on the transducer linkers for better clarity. The transducer-TOPRIM linkers are now better displayed in the density.

The linker between the ATPase and the TOPRIM domains was clearly visible in a crystal structure of *M. tuberculosis* DNA gyrase. Are they very different?

In the crystal structure mentioned by the reviewer, the DNA-free *M. tuberculosis* DNA gyrase adopts an open conformation. In this configuration, the transducer-TOPRIM linkers which are also in this case flexible loops, are indeed visible, but laying on the TOPRIM domain in absence of DNA.

In the *E. coli* complex we present here, these linkers adopt the position that corresponds to a dimerized form of gyrase in presence of the DNA G-fragment, a conformation that represents an important functional step of the catalytic cycle. This is what we meant in the result section concerning the transducer when mentioning “The loops connecting the ATPase domain and DNA-binding/cleavage domain are now clearly built and positioned in the DNA-bound gyrase model”.

To make it more explicit, we have modified this paragraph to mention the DNA-free open *Mt* gyrase structure by comparison. In addition, we have moved this part in the second paragraph of the result section mentioning the completeness of the model, to make the description of the structure more coherent:

“The resulting model now shows clearly the details of the intertwined structure of the DNA gyrase heterotetramer (Fig. 2b). The dimeric ATPase domain is sitting in an orthogonal orientation above the DNA-binding/cleavage domain (~95°) as previously observed with the *T. thermophilus* DNA gyrase¹² (Supplementary Fig. 6a). The density of the flexible loops connecting the ATPase domain to the TOPRIM domain are well defined allowing to position these elements crossing on top of the G-segment in the DNA-bound gyrase conformation (Fig. 2d). In the DNA-free open structure of the *M. tuberculosis* DNA gyrase, these linkers are laying apart on the surface of the TOPRIM domain³³.”

We added the corresponding reference in the reference list:

33. Petrella, S. *et al.* Overall Structures of Mycobacterium tuberculosis DNA Gyrase Reveal the Role of a Corynebacteriales GyrB-Specific Insert in ATPase Activity. *Structure* 27, 579-589 e5 (2019).

5. Concerning the explanation as to why the *E. coli* mutant R286Q has a reduced negative supercoiling activity, the argument is indirect, using the fact that the *Thermus* transducer is stabilized by a hydrophobic core involving 5 residues. To me, this argument sounds not totally convincing. These 5 residues are also hydrophobic in *E. coli* (L, W, F, M and A).

We have analyzed again both transducer structures. The hydrophobicity of this patch is indeed conserved but 3 aliphatic side chains in *E. coli*, V372, M289, L256, are replaced by aromatic residues Y370, Y287, F254 in *T. Thermophilus*. We believe that these bulky side chains could possibly confere additional stability to the transducer. In addition, a hydrogen bond between H256 and Y287 in *Thermus thermophilus* is possibly compensating for the absence of the salt bridge. We have added this information in the discussion:

“In *T. thermophilus*, the equivalent position K284 is pointing in the cavity and could potentially mediate an interaction with DNA, however our experiments show that point mutations have no effect on the catalytic activities In contrast, the analysis of the *E. coli* structure shows that R286 is engaged in an interaction network with E264 and R316 anchoring together secondary structures of the transducer and providing rigidity to the transducer terminal helix (Fig. 4b). In *T. thermophilus*, the transducer is instead stabilized by a hydrophobic core where aliphatic side chains are replaced by aromatic residues and a different hydrogen bond network, notably involving H256 and Y287 (Fig. 4a).

Rather than a direct effect involving DNA contact, we conclude that efficient allosteric transmission of ATP hydrolysis relies on the transducer rigidity that is maintained by species-specific interaction networks.”

We have also revised accordingly former Fig. 5 (now Fig. 4) and supplementary Fig. 8 and 9.

The equivalent position in *T. thermophilus* is Ala264 (not stated in the text), corresponding to a loss of the salt bridge. Could it be possible to analyze the mutant Ec E264A, in order to mimic the *T. thermophilus* enzyme at this position and to confirm this hypothesis? *T. thermophilus* is the only sequence in the alignment that possesses an alanine a position 264.

The goal if this manuscript was not to analyze the *Thermus* enzyme but rather to determine the *E. coli* gyrase architecture. We however thought it would be interesting to compare the 2 enzymes for the following reasons:

- Tingey and Maxwell analyzed in 1996 the role of *E. coli* R286, based on a crystal structure and published by the team of D. Wigley in 1991, but for which the atomic coordinates were not released. The article proposed a figure displaying the side chain of R286 pointing into the transducer cavity. Tingey and Maxwell's study concluded that this residue is directly involved in T-segment capture.

- Looking backwards at this study, this conclusion seemed puzzling since the analysis of the crystal structure of *E. coli* ATPase domain published in 2000 by Brino *et al.* (PDB:1Ei1), showed that this residue is engaged in a hydrogen bond network and is not pointing into the transducer cavity.

- Indeed the *T. thermophilus* gyrase has a lysine at this position which could perform the same function but has an alanine instead of an aspartic acid at position 264 (corrected to 262), excluding a salt bridge.

Therefore, *T. thermophilus* provides an alternate sequence for an enzyme performing the same activity and presenting an overall same architecture.

- Due to the importance of the transducer domain in the enzyme and allosteric mechanism as emphasized by the overall structure we provide, we decided to revisit these experiments by performing the exact same mutations as Tingey and Maxwell to clarify this point.

We have now made this point more explicit in the result section:

“Based on the first crystal structure of the ATPase domain⁸, it was initially suggested that residue R286 in *E. coli* (Ec) points towards the cavity of the transducer domain and contributes to DNA capture and therefore is a key element of the enzyme's allostery³⁷.”

To describe more completely the hydrogen bond network, we have added the information that this salt bridge is also forming hydrogen bonds with R316 and added a new figure:

“However, further analysis of this area from the crystal structure of Brino *et al* shows that *Ec* R286 is engaged in a salt bridge with E264, in a hydrogen bond network with R316 ³² (Fig. 4b)”.

...

“*Th* K284 at the same position points toward the N-gate central cavity and is not engaged in a hydrogen bond network since E264 is replaced by an A262 and R316 by a L316 ³⁷. This organism therefore presents an alternative sequence to *E. coli* in this region with a conserved folding and accomplishing the same set of catalytic activities.”

Although we felt that mutation of R286 would be sufficient to clarify the role of this residue in the structure-function analysis of the structure, we, nevertheless, proceeded to the *E. coli* GyrB E264A mutation to mimic the *Th* A262 position. The proteins were purified and all assays were performed with the same conditions and controls as for the other mutants. This analysis required several weeks of experiments and the involvement of another team member that we have now included in the authors list.

The mutation of E264 indeed alters both the ATPase and DNA supercoiling activities with a stronger effect than the R286Q mutation, confirming the importance of this salt bridge. The strong impact of the E264A mutation on the activities can also be explained by an additional interaction with R316 that is now showed in Fig. 4b.

All related figures were revised to include the new data and the information has been added in the result section together with the other mutants.

Thermus aquaticus possesses a threonine. Is T. thermophilus an exception? What about other Deinococcus-Thermus bacteria?

We performed another multiple sequence alignment on this region to answer this question (see below). The *Thermus* species possess either an alanine or a threonine at this position whereas *Deinococcus* species have an aspartate or serine.

These obvious compensatory networks are established depending on the species sequence since more than one interaction contributes to the rigidity of the structure in this area.

We did not include the additional information in the manuscript since we think a detailed sequence analysis would be beyond the scope of this study that is primarily dedicated to the full architecture of the enzyme. We are indeed aware that this structural data now opens many questions that can be the starting point for new investigations.

I would also simplify the paragraph that discusses this point in the discussion section. The discussion concerning the lysine/arginine replacement is not really useful with respect to the importance of the loss of the salt bridge.

We have now completely reorganized the result/discussion paragraphs on this point. Extensive explanations that were inserted in the results section have been transferred to the discussion where more emphasis was put on the salt bridge of *E. coli*.

One additional remark, the numbering of these residues seems mixed up. Why Lys284 in *T. thermophilus* and not Lys286? Based on the alignment, there is no insertion/deletion between *E. coli* and *T. thermophiles*.

The alignment in Supplementary Fig. 8c is restricted to a portion of the ATPase domain sequence to highlight the discussed positions. In this region, there is indeed no gap between the 2 sequences. However, when considering the peptide sequence from residue 1 to 286, according to the sequence files, the *T. Thermophilus* GyrB subunit (UniProtKB: Q5SHZ4 GYRB_THET8) has 2 residues less than the *E. coli* protein (UniProtKB - P0AES6 GYRB_ECOLI) in this region, explaining the difference in numbering. For the same reason we have corrected all the numbering, A264 is now A262 in *Thermus*.

6. In the discussion section, I do not totally agree with the statement “For decades, the opening of the DNA-gate was predicted to follow a “book-opening” motion, in which the two halves of the DNA-gate simply remain along the axis of G-segment 2,3,41. However, it has been shown recently that the DNA-gate opening mechanism is more subtle and is achieved by a “sliding and swiveling” motion”. Even if the animation presented in the movie published by Berger in 1996 shows a simple “book-opening” motion, the structure of the breakage-reunion domain of the *S. cerevisiae* topoisomerase II (PDB code 1BGW, Berger et al., 1996) clearly indicates that the motion is closer to a screwing motion following an elliptic path, similar to the sliding and swiveling motion. We could confirm this by calculations of the transition path we performed using the Path Optimization and Exploration approach (in collaboration with Arnaud Blondel, Institut Pasteur, unpublished results). I would recommend reviewing this section.

This comment was addressed to the general reader outside of the topoisomerase field to insist on the sophistication of the mechanism that is indeed difficult to represent in classical 2D representation or with a morphing between states. However, we simplified and rephrased this section in order to be more accurate with regards to the description of the available literature as requested by the reviewer:

“During T-segment strand passage, the DNA-binding/cleavage domain needs to undergo several conformational changes. Crystal structures of DNA-binding/cleavage domains in different states have led to propose that the DNA-gate opening is achieved by a “sliding and swiveling” motion of the two halves against each other, breaking the G-segment axis^{2,3,39,44}.”

7. Furthering the topic of the conformational changes of the DNA-binding and cleavage domain during opening of the DNA-gate, the authors present two reconstruction corresponding to two states, a closed and a pre-opening state. In Fig. 6a, it is difficult to apprehend the pre-opening state. Was the superimposition performed on the grey monomers? Fig. 6b should help to better clarify this point, but the structural differences in this part of the figure are also difficult to see.

We agree that this movement is not easy to represent in a 2D figure, this is why we also provided a Supplementary Movie 2. To improve this figure, we propose a new panel a. in former Fig. 6 now renumbered Fig. 5. We hope this helps clarifying the movements of the domains. We also rephrased the legend to better explain how the superimposition was conducted.

In addition, we cannot neglect the fact that, if one takes into account all the structural data concerning the DNA-binding and cleavage domain of type II topoisomerases, including structures of gyrases, topo IV and topo II, more than 50 structures are available. Even if it would be a huge work to analyse all these structures, did the authors check if there is a structure that could be assimilated to a pre-opening conformation?

Some comparison of the spacing between the catalytic tyrosines is provided in Supplementary Fig. 10 that allows to compare the opening of the DNA-gate. The shorter is the distance between the tyrosines, the larger is the DNA-gate opening. Our pre-opening conformation is more “open” than the closed conformation and only in a “pre-opening” configuration when compared with the opened conformation observed in the PDB 5ZEN.

We have not included all the superimpositions we performed in the present article, we rather focused on the connections between the domains which are important for the allosteric control of the enzyme activities. We agree an extensive comparison to all available PDB would be interesting but in the context of a review study, which is out of the scope of the present work. As a matter of fact, as we were revising this manuscript, we came across the publication of a review in Journal of Molecular Biology by Bax *et al.* online July 10th:
<https://doi.org/10.1016/j.jmb.2019.07.008>

For your information, we have superimposed our DNA-binding/cleavage domain models to several crystal structures from human and bacterial species obtained in presence of different drugs. A first analysis shows that our 2 states present the smallest rmsd with PDB: 2XCT that was also in complex with a double-nicked DNA (Bax *et al*, Nature 2010). The closed conformation is closer to this state compared to the pre-opening conformation. A detailed study would be required to analyze these details further.

8. When comparing to the *T. thermophilus* cryo-EM structure, the authors propose that the *E. coli* conformation precedes the conformation of the *T. thermophiles* one. This is highly hypothetic. A bending angle of the segment of 150° should be energetically unfavorable.

We agree that comparison of the *T. thermophilus* and *E. coli* conformations is uncertain given the lower resolution of the cryo-EM reconstruction of the *T. thermophilus* gyrase. This comment being also raised by reviewer 3, we removed the *T. thermophilus* complex from Fig. 4, transferred the geometry analysis to Supplementary Fig. 7 and simplified this discussion. The paragraph title has now been changed to “Analysis of the overall conformation of the *E. coli* DNA gyrase DNA complex.”

We mentioned the *T. Thermophilus* complex but removed the extensive comparison between the 2 structures.

As a side remark, how are the -10 and 10 degrees angle measured?

The -10 and +10° angles were measured respective to the plan of the DNA-binding groove. We have added this information in the legend of Supplementary Fig. 6.

In addition, the sentence « Interestingly, a recent single-molecule study described a Ω state, possibly similar to the *E. coli* complex and occurring before the α state in the catalytic cycle, more reminiscent of the *T. thermophilus* conformation” is difficult to follow. I would recommend clarifying this sentence.

We have now reformulated this sentence to take into account the modifications of the whole paragraph:

“Interestingly, a recent single-molecule study identified a Ω conformational state, possibly similar to the *E. coli* complex and preceding the α state in the catalytic cycle, a conformation favorable to the T-segment strand passage¹³. This transition would require a different orientation of the pinwheels which motion might be controlled and regulated by the acidic CTD tail in contact with the DNA-gate as previously suggested⁴¹.”

They also mentioned the acidic extreme-CTD tail of the *E. coli* enzyme. Is it visible in the density? What about the CTD tail of *T. thermophilus*?

All densities that were visible were interpreted and reported in the figures. The flexible CTD tail is so far not visible in any of the available cryo-EM, or crystal structures of *E. coli* gyrase. The CTD of *T. Thermophilus* does not possess this acidic tail and only ends with 3 glutamic acids (see alignment).

```
sp|Q5SIL4|GYRA_THET8          TKVGRLAALLKVRGGEDLLVLSRRGLAIRTPVAEIRQYSRATAGVRVMNLPEDDEVASAF 799
1ZI0:A|PDBID|CHAIN|SEQUENCE  ERNGLVVGAVQVDDCDQIMMITDAGTLVRTRVSEISIVGRNTQGVILIRTAEDENVVGLQ 303
sp|P0AES4|GYRA_ECOLI         ERNGLVVGAVQVDDCDQIMMITDAGTLVRTRVSEISIVGRNTQGVILIRTAEDENVVGLQ 837

sp|Q5SIL4|GYRA_THET8          VVEEEK----- 805
1ZI0:A|PDBID|CHAIN|SEQUENCE  RVAE----- 307
sp|P0AES4|GYRA_ECOLI         RVAEPVDEEDLDTIDGSAEGDDEIAPEVDVDDEPEEE 875
```

10. The figures could be slightly improved.

- The inset of figure 2 has already been mentioned

See answer to question 3.

- Color code of the CTD in figure 3 has already been mentioned

See answer to question 3.

- In figure 4, the labeling should be improved (placement of the value angle labels, space before Angström, ...)

This figure has been removed and part of the panels were transferred to Supplementary Fig. 6.

- The shading in figure 5d is missing, rendering it flat

This figure has been revised to better display the interaction network in the transducer of both species.

- The clarity of figure 6 has already been mentioned

See answer to question 7.

- In Figure S8, the position in yellow is totally impossible to read

This figure has been revised to better highlight the positions in the alignment.

11. Some titles are misleading or inaccurate, especially the one of the last “Results” section that deals with drug and magnesium binding. I would recommend going through all the titles.

This title has now been changed to “Identification of the antibiotic molecule and ions in the 4.0 Å structure of the gyrase DNA-binding/cleavage domain”, to be more explicit.

In the result section, we also changed the following titles:

- from “Analysis of conserved structural elements of the transducer domain involved in allosteric control” to “Structural and functional analysis of conserved structural elements in the transducer domain”

- from “Analysis of the DNA-gate conformations” to “Deconvolution of the DNA-gate conformations”

In the Discussion section, we changed the following titles:

- from “Analysis of the overall conformation of the *E. coli* DNA gyrase DNA complex” to “Gepotidacin traps a G-segment wrapping conformation of *E. coli* DNA gyrase”

- from “Position of the β -pinwheel and GyrA-box” to “Position of the β -pinwheel and GyrA-box in DNA wrapping”

Minor remarks:

- P3. ADPNP is not the only non-hydrolysable analog of ATP

We agree with the reviewer. We have rephrased the sentence: “which was further stabilized using ADPNP, a non-hydrolysable analog of ATP”

- P4. “Defined alpha helices”: which ones?

This sentence was to emphasize that the quality of the data is good enough to observe high-resolution features, such as alpha helices and DNA duplex, already at the 2D classification step. From the qualitative analysis of these images it is not possible to assign them precisely. This is only possible at the level of the analysis of the 3D reconstruction. The most defined density of

alpha helices was observed in the DNA-binding/cleavage domain where the local resolution is the best.

- P5. “GyrB TOPRIM insertion relative to the catalytic domains”. The catalytic domain is composed of the TOPRIM domain and the N-terminal domain of GyrA (which has several names in the literature), I recommend to reformulate this sentence (see also last point of the minor remarks)

The DNA-binding/cleavage domain can also be designated as catalytic domain. To avoid misinterpretation, we have modified “catalytic domain” by “DNA-binding/cleavage domain”.

The TOPRIM insertion is part of the DNA-binding/cleavage domain. However, it protrudes outside of the catalytic domain and was found to adopt different positions relative the rest of the catalytic domain. We have now specified “... in two different states with different positioning of the GyrB TOPRIM insertion relative to the rest of the DNA-binding/cleavage domain”.

- P6. “the atomics models of the GyrA and GyrB domains” Precise *E. coli*

This has now been specified in the text.

- ATPase head: whole or part of the ATPase domain? Please, clarify

This term was misleading. We have replaced it by ATPase domain, which comprises the GHKL and Transducer subdomains.

- Legend of figure 3: a space is missing (130bp)

We have corrected this mistake.

- With the growing number of sequenced genomes, it would be very interesting to mention to what extent this insertion exists in bacterial gyrases. Is it found in all gyrases of the *Escherichia* genus? The Enterobacteriaceae family? The Enterobacterales order?

We thank the reviewer to bring this interesting question to our attention. In the original structural study on the TOPRIM insertion performed by Schoeffler *et al.* in 2010⁷, the authors have provided a multiple sequence alignment showing that the proteobacteria (which comprises the enterobacteria), the acidobacteria, the chlamydia and the aquificales possess the TOPRIM insertion.

We also performed a multiple sequence alignment on this region using a broader set of organisms covering the whole bacteria kingdom (see below). We also found that the organisms harboring the TOPRIM insertion belong to these bacterial phyla. This point being raised in the study of Schoeffler *et al*, we did not elaborate on this point in the manuscript.

11. Schoeffler AJ, May AP, Berger JM. (2010) A domain insertion in Escherichia coli GyrB adopts a novel fold that plays a critical role in gyrase function. *Nucleic Acids Res.* 38(21):7830-44.

- What is included in the term “DNA-binding/cleavage domain”? The N-terminal domain of GyrA? The N-terminal domain of GyrA and the TOPRIM domain”. The literature is highly inconsistent with respect to this terminology. It would be worse to clarify it.

The DNA-binding/cleavage domain comprises the N-terminal part of GyrA (residues 1-524) and the C-terminal part of GyrB (residues 402-804). More precisely, it is composed of 2 copies of the following subdomains: TOPRIM (GyrB), WHD (GyrA), Tower (GyrA), and Coiled-coil (GyrA). The figure below has been inserted in Fig. 2a. It is also added in Supplementary Fig. 13.

Signed,
Claudine Mayer

Reviewer #3 (Remarks to the Author):

Broeck et al. present structure of a complete Gyrase-DNA complex from *E. coli* using phase-plate single particle cryo-EM, revealing the overall architecture of this extremely flexible and essential DNA topoisomerase at great details for the first time. This is beautiful work and a big advance in the field. In particular, the galleries of structures generated from this study revealed an allosteric movement of the complex by comparing the closed versus the pre-opening states. Furthermore, the authors were able to identify the antibiotic location, thanks to their near atomic resolution density map produced from state-of-the-art data collection and analysis. The manuscript is well written and they have done a good job selecting figures that illustrate important features of the structure models. All details leading to the cryo-EM reconstructions, models and fittings are very clearly described. There is simply nothing to be complained about the technical approaches chosen by the authors. The results reported here will guide the next few years of advances in the study of DNA topoisomerase; once published this work seems destined to become a classic.

We thank the reviewer for the positive appreciation of our work and for the time taken to review our manuscript. We have responded to reviewer's comments below.

I have only a minor comment that might guide improvement during preparation of a final version of the manuscript. In figure 4, the authors attempted to compare the architecture of DNA Gyrase complex and the associated DNA topology from *E. coli* (this study) and *T. Thermophilus*. However, the cryo-EM structure of the *Thermophilus* Gyrase was determined prior to the resolution revolution and it was limited to 23 Å-16.9 Å at the time. At this resolution, it's going to be hard to confidently delineate the path of the DNA. Indeed, in the rigid body docking, there was clear empty density in the *Thermophilus* structure which could fit the G-segment in a similar way as in the *E. coli* structure. Unless the authors can present higher resolution maps for the *Thermophilus* Gyrase structure, it would be wise to not put such an emphasis on the distinct mode of DNA engagement and bending between the two similar systems. In summary, this is an impressive piece of work.

We agree that comparison of the *T. thermophilus* and *E. coli* conformations is uncertain given the lower resolution of the cryo-EM reconstruction of the *T. thermophilus* gyrase. This comment being also raised by reviewer 2, we removed the comparison to *T. thermophilus* in Fig. 4 and simplified the discussion of this part in the first paragraph of the discussion entitled "Gepotidacin traps a G-segment wrapping conformation of *E. coli* DNA gyrase".

REVIEWERS' COMMENTS:

Reviewer #2 (Remarks to the Author):

The revised version of the manuscript of Vanden Broeck et al. has addressed all the concerns from the previous review and is now suitable for publication